# MASS-DPO: Multi-negative Active Sample Selection for Direct Policy Optimization

## Abstract

Multi-negative preference optimization under the Plackett–Luce (PL) model extends Direct Preference Optimization (DPO) by leveraging comparative signals across one preferred and multiple rejected responses. However, optimizing over large pools of negatives is computationally prohibitive, and many candidates contribute redundant gradients due to their similar effects on policy updates. To address this, we introduce **MASS-DPO**, which derives the Fisher information matrix directly from the PL objective and shows that the problem of selecting negatives naturally reduces to a D-optimal design formulation. This formulation guarantees maximal informativeness and comprehensive coverage of the current policy's weaknesses. Moreover, the log-determinant criterion underlying D-optimal design admits a submodular structure, which we exploit through an incremental greedy algorithm that provides the natural computational realization of D-optimality, combining scalability with theoretical rigor. This incremental greedy strategy efficiently resolves the combinatorial complexity inherent in selecting a D-optimal negative set from large candidate pools. We establish convergence guarantees and finite-sample error bounds under this framework, and empirically demonstrate that MASS-DPO improves optimization efficiency and enhances downstream performance, achieving stronger alignment with substantially fewer negatives.

## 1 Introduction

Direct Preference Optimization (DPO) (Rafailov et al., 2023) aligns models directly with human preferences by optimizing pairwise comparisons without explicitly constructing reward functions (Christiano et al., 2017; Ouyang et al., 2022; Stiennon et al., 2020b). Recent works have generalized DPO using the Plackett-Luce (PL) model (Plackett, 1975; Luce et al., 1959) to accommodate multiple negative samples, enriching preference signals for more robust alignment. However, current multi-negative approaches such as Softmax-DPO (S-DPO) (Chen et al., 2024) and Direct Multi-Preference Optimization (DMPO) (Bai et al., 2024)typically select negatives randomly or heuristically, leading to redundant gradient information and computational inefficiencies.

To address these limitations, we propose MASS-DPO (**M**ulti-negative **A**ctive **S**ample **S**election for **D**irect **P**reference **O**ptimization), a theoretically-grounded active negative selection setting derived from the multi-negative Plackett-Luce preference optimization objective (Plackett, 1975; Luce et al., 1959). MASS-DPO formulates negative sample selection as a D-optimal design problem (Pukelsheim, 2006; Kiefer, 1959), leveraging the Fisher information matrix to measure the informativeness of each negative candidate (Fisher & Russell, 1922; Chaloner & Verdinelli, 1995; Flaherty et al., 2005; Kirsch et al., 2019). This selection is non-trivial, as many negative responses produce highly similar gradients under the PL objective, resulting in overlapping optimization signals that fail to introduce novel information for improving the policy. Without careful selection, the model repeatedly updates toward already-learned directions, leading to inefficient learning and slower convergence. To overcome this, MASS-DPO actively selects a diverse and informative subset of negatives by optimizing for coverage and signal diversity through a D-optimal design formulation.

Although the D-optimal formulation provides theoretical guarantees of optimal negative sampling, it introduces significant combinatorial complexity when selecting from a large candidate pool (Krause & Guestrin, 2012; Kirsch et al., 2019). To efficiently overcome this challenge, we further propose an incremental greedy algorithm that efficiently identifies a compact subset of negatives, prov-

ably equivalent in optimality to the full combinatorial selection (Nemhauser et al., 1978; Krause & Guestrin, 2012; Kirsch et al., 2019). This incremental approach effectively balances theoretical rigor with computational practicality, aligning with previous successful applications of greedy algorithms in information-theoretic sample selection (Sener & Savarese, 2017; Kirsch et al., 2019; Kveton et al., 2025).

We provide comprehensive theoretical analyses, establishing finite-sample estimation error bounds and convergence guarantees under our proposed selection framework. Empirically, we demonstrate that MASS-DPO significantly enhances optimization efficiency and alignment quality, achieving superior downstream task performance using substantially fewer negatives compared to existing methods across diverse benchmarks in language modeling and recommendation tasks. We summarize our contributions as follows:

- We propose MASS-DPO, an active negative sample selection method formulated as a D-optimal design problem, theoretically derived from the multi-negative Plackett-Luce optimization objective.

- We further introduce an incremental greedy selection algorithm, ensuring theoretical equivalence to the global optimal solution while significantly reducing computational overhead.

- We establish rigorous theoretical guarantees, including finite-sample estimation error bounds and convergence properties for MASS-DPO.

- Extensive empirical evaluations demonstrate that MASS-DPO outperforms baseline methods in optimization efficiency and downstream task performance across multiple language models and both recommendation and multiple-choice QA tasks.

## 2 RELATED WORKS

**Direct Preference Optimization.** DPO (Rafailov et al., 2023) aligns language models with human preferences by optimizing likelihood ratios of preferred over dispreferred responses, avoiding explicit reward modeling and associated complexities such as reward misgeneralization seen in RLHF (Christiano et al., 2017; Ouyang et al., 2022; Stiennon et al., 2020b). Recent extensions, such as introducing dynamic margins in ODPO (Amini et al., 2024) and computational optimizations via prefix sharing (Wang & Hegde, 2024), have further improved DPO's effectiveness and efficiency. However, standard DPO methods are typically restricted to binary preference pairs, which limits the diversity of supervision and often results in inefficient use of available preference data. In contrast, our approach extends beyond binary comparisons by leveraging actively selected, informative multi-negative samples, enabling more efficient and robust alignment.

**Multi-negative Preference Optimization.** Recent work has extended standard DPO's binary preference pairs to leverage multiple negatives for richer comparative signals and enhanced alignment. Softmax-DPO (S-DPO) (Chen et al., 2024) generalizes the pairwise Bradley–Terry loss (Bradley & Terry, 1952) to Plackett–Luce ranking (Plackett, 1975; Luce et al., 1959), providing richer gradient signals. Direct Multi-Preference Optimization (DMPO) (Bai et al., 2024) averages over multiple negatives to promote diverse negative learning. Multi Pair-wise Preference Optimization (MPPO) (Xie et al., 2024) extends DPO by directly modeling multi-negative feedback with average-likelihood loss, removing the need for a reference model and enabling flexible use of negative samples. Tree Preference Optimization (TPO) (Liao et al., 2024) structures multi-negative alignment through hierarchical preference decomposition. Despite these advances in multi-negative preference optimization, current methods still largely depend on heuristic or random negative selection strategies. Our work addresses this limitation by proposing MASS-DPO, which leverages D-optimal design for theoretically grounded, strategic negative sample selection.

## 3 PRELIMINARIES

### 3.1 DIRECT PREFERENCE OPTIMIZATION

Direct Preference Optimization (DPO) (Rafailov et al., 2023) aligns a learned policy with human pairwise judgments (Christiano et al., 2017; Stiennon et al., 2020a; Ouyang et al., 2022) without

an explicit reward model. Under the Bradley-Terry-Luce framework (Bradley & Terry, 1952), two responses $y_1, y_2$ to prompt $x$ with latent scores $r(x, y_1), r(x, y_2)$ satisfy

$$p^*(y_1 \succ y_2 \mid x) = \sigma(r(x, y_1) - r(x, y_2)), \tag{1}$$

where $\sigma(z) = 1/(1 + e^{-z})$. Rearranging the optimal-policy relation gives an implicit reward decomposition up to an additive normalizer $Z(x)$:

$$r(x, y) = \beta \, \log\frac{\pi^*(y \mid x)}{\pi_{\text{ref}}(y \mid x)} + Z(x), \quad Z(x) = \sum_y \pi_{\text{ref}}(y \mid x) \cdot \exp\!\Big(\tfrac{1}{\beta}\, r(x, y)\Big) \tag{2}$$

Substituting equation 2 into equation 1 and simplifying leads to the DPO training objective

$$\mathcal{L}_{\text{DPO}}(\theta) = -\mathbb{E}_{(x, y_1, y_2) \sim D}\Big[\log \sigma\Big(\beta \, \log\frac{\pi_\theta(y_1 \mid x)}{\pi_{\text{ref}}(y_1 \mid x)} - \log\frac{\pi_\theta(y_2 \mid x)}{\pi_{\text{ref}}(y_2 \mid x)}\Big)\Big]. \tag{3}$$

### 3.2 MULTI-NEGATIVE PREFERENCE OPTIMIZATION

Multi-negative preference optimization generalizes the Direct Preference Optimization framework (Rafailov et al., 2023) to better align language models with multiple negative preferences. While traditional DPO employs the Bradley-Terry (BT) model (Bradley & Terry, 1952) to capture pairwise comparisons, multi-negative preference optimization leverages the Plackett-Luce (PL) model (Plackett, 1975; Luce et al., 1959) to accommodate the ranking of a preferred item against multiple disfavored items.

Consider a user prompt $x_u$ that is formed from historical interactions, along with a preferred item $e_p$ and a set of dispreferred items $E_d$. The aim is to maximize the probability that the preferred item $e_p$ is ranked above every item in $E_d$, as described by

$$p^*(e_p \succ E_d \mid x_u) = \frac{\exp(r(x_u, e_p))}{\sum_{e_d \in \{e_p\} \cup E_d} \exp(r(x_u, e_d))}, \tag{4}$$

where $r(x_u, e)$ is the latent reward function defined over the prompt-response pairs in the RLHF framework (Ouyang et al., 2022). From Eq. equation 4, we obtain the following multi-negative preference loss:

$$\mathcal{L}(\theta) = -\mathbb{E}_{(x_u, e_p, E_d) \sim D}\Big[\log \sigma\Big(-\log \sum_{e_d \in E_d} \exp\big(\beta \, \Delta(x_u, e_d, e_p)\big)\Big)\Big], \tag{5}$$

with $\sigma(\cdot)$ denoting the sigmoid function and $\Delta(x_u, e_d, e_p) = \log\frac{\pi_\theta(e_d \mid x_u)}{\pi_{\text{ref}}(e_d \mid x_u)} - \log\frac{\pi_\theta(e_p \mid x_u)}{\pi_{\text{ref}}(e_p \mid x_u)}$. Notably, this formulation reverts to the original DPO setup when the set of dispreferred items contains just a single element (i.e., $|E_d| = 1$). This naturally extends DPO by incorporating multi-negative preference alignment into language model training for recommendation tasks.

## 4 MASS-DPO: MULTI-NEGATIVE ACTIVE SAMPLE SELECTION

In multi-negative preference optimization tasks (*e.g.*, recommendation, multiple-choice QA, information retrieval), the selection of negative samples significantly influences alignment efficiency and effectiveness. *Uninformative* negatives, already well-separated from preferred responses, waste gradient computations and hinder convergence (Yang et al., 2023; Kalantidis et al., 2020; Robinson et al., 2020; Zhang et al., 2022). Thus, the key challenge is strategically selecting a compact yet informative subset of negatives to highlight the policy's weaknesses while maintaining numerical stability (Ma et al., 2024; Kirsch & Gal, 2022; Fan et al., 2023). To address this, we propose MASS-DPO (Figure 1), an active negative selection method formulated as a D-optimal design problem (Pukelsheim, 2006; Cohn, 1993; Kirsch et al., 2019), maximizing a Fisher-information surrogate (Fisher & Russell, 1922; Jung & Lee, 2021; Liu et al., 2024; Neilsen et al., 2018; Sourati et al., 2017; Chaloner & Verdinelli, 1995; Ash et al., 2021). By maximizing this surrogate, MASS-DPO effectively minimizes the volume of the confidence ellipsoid of policy parameters connecting computational efficiency with robust statistical guarantees (Sec. 5). We outline the core assumptions and derive gradient and curvature expressions central to our analysis and optimization approach.

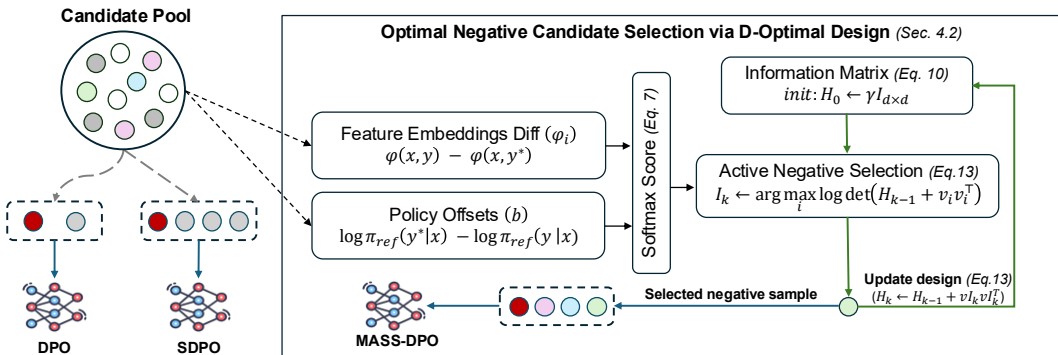

Figure 1: Overview of MASS-DPO's D-optimal selection. Each candidate is scored by its feature-difference and policy offset via softmax (Equation (8)). The green loop denotes the incremental greedy update: starting from $H_0$, we incrementally pick the negative that maximally increases $\log \det H$, then update $H$ accordingly until $n$ samples are selected.

## 4.1 SETTING

Following (Kveton et al., 2025; Riquelme et al., 2018; Das et al., 2024; Mukherjee et al., 2024; Liu et al., 2024; Thekumparampil et al., 2024) in regret minimization and reward-model active learning, we linearize the policy's final layer to obtain a tractable Fisher-information objective. We assume:

**Assumption 4.1.** *To enable tractable analysis and algorithmic design, we assume the policy under consideration takes a log-linear form:*

$$\pi(y \mid x; \theta) \propto \exp\big(\phi(x, y)^\top \theta\big), \tag{6}$$

*where $\phi(x, y) \in \mathbb{R}^d$ denotes the feature embedding of the context–response pair $(x, y)$, and $\theta \in \mathbb{R}^d$ the model parameters.*

Under Assumption 4.1, we can represent the relative preference between a preferred response $y^*$ and a set of negative responses $\{y_i\}_{i=1}^n$ through feature differences. Specifically, defining

$$\phi_i = \phi(x, y_i) - \phi(x, y^*), \quad b_i = \log \frac{\pi_{\text{ref}}(y^* \mid x)}{\pi_{\text{ref}}(y_i \mid x)},$$

allows us to write the multi-negative DPO loss compactly in terms of the log-sum-exp operator.

$$L(\theta; S_n) = -\log \sigma\Big(-\log \sum_{i \in S_n} \exp\big(\beta\,(\phi_i^\top \theta + b_i)\big)\Big), \tag{7}$$

where $S_n = \{y_1, \ldots, y_n\}$ is the chosen negative set, $\sigma(\cdot)$ the sigmoid. Then, we propose the following lemmas to quantify how each candidate negative alters the gradient and curvature. These statistics show that negatives whose feature differences are *diverse* and *orthogonal*, enlarge the information matrix the most, while redundant examples leave their volume almost unchanged.

**Lemma 4.1.** *Definition of auxiliary terms: normalization factor $Z_n$ and softmax weights $p_j$ for gradient computation*

$$p_j = \frac{\exp\big[\beta(\phi_j^\top \theta + b_j)\big]}{\sum_{k \in S_n} \exp\big[\beta(\phi_k^\top \theta + b_k)\big]}, \quad Z_n = -\log \sum_{i \in S_n} \exp\big[\beta(\phi_i^\top \theta + b_i)\big] \tag{8}$$

*Then the gradient of equation 7 with respect to $\theta$ is given by*

$$\nabla_\theta L(\theta; S_n) = \beta\,(1 - \sigma(Z_n)) \sum_{j \in S_n} p_j\,\phi_j. \tag{9}$$

The detailed derivation is provided in Appendix A.1. This result shows that the gradient is a weighted combination of feature differences, scaled by the probability of misranking, thus facilitating intuitive interpretations in terms of correction signals. We observe that $p_j$ emphasises negatives whose score margin is small, which are borderline mistakes that have the greatest influence on the policy, corroborating the need to focus selection on *hard yet informative* examples.

**Lemma 4.2.** *Let $\phi = \sum_{j \in S_n} p_j \phi_j$ denote the expected feature difference under the softmax distribution. The Hessian of equation 7 is then*

$$\nabla^2 L(\theta; S_n) = \beta^2 (1 - \sigma(Z_n)) \Big[ \sigma(Z_n) \phi \phi^\top + \sum_{j=1}^n p_j (\phi_j - \phi)(\phi_j - \phi)^\top \Big]$$

$$\succeq \beta^2 (1 - \sigma(Z_n)) \sum_{j=1}^n p_j (\phi_j - \phi)(\phi_j - \phi)^\top, \tag{10}$$

*The inequality follows because the dropped term $\sigma(Z_n) \phi \phi^\top$ is positive semidefinite, hence removing it yields a valid Loewner lower bound which is positive semi-definite and captures both the low-rank and dispersion contributions to curvature. The detailed derivation is provided in Appendix A.2.*

Based on the lower bound of the Hessian matrix of the multi-negative DPO objective, we directly maximize the latter motivates a determinant objective that prefers sets to spread along orthogonal directions in feature space. These gradient and Hessian expressions form the basis for our multi-negative active sampling strategies, enabling principled optimization of negative sets under budget constraints while controlling estimation uncertainty and convergence behavior.

## 4.2 NEGATIVE SELECTION VIA D-OPTIMAL DESIGN

When selecting from a large-scale negative pool in multi-negative DPO, more negatives can improve parameter estimates, but those samples can add little beyond what is already conveyed by a smaller, well-chosen subset. MASS-DPO enables negative selection as a *D-optimal design* (Kiefer, 1959; Pukelsheim, 2006; Kirsch et al., 2019) problem that explicitly maximizes the information gain (Chaloner & Verdinelli, 1995) about the policy parameters.

**Fisher-information objective.** For a candidate negative $j \in \mathcal{D}$ let $v_j = \sqrt{p_j} (\phi_j - \phi)$ be its Fisher-information contribution, where $p_j$ is the softmax weight derived in Section 4.1. Given a subset $S \subseteq \mathcal{D}$ we define the regularised information matrix

$$H(S) = \gamma I + \beta^2 (1 - \sigma(Z_n)) \sum_{j \in S} v_j v_j^\top, \quad \gamma > 0. \tag{11}$$

We initialize the design matrix with a ridge term $\gamma I_d$, where $\gamma > 0$ is a constant regularizer that guarantees $H(S)$ is well defined and numerically stable for all subsets $S$, and $\beta$ is a hyperparameter that controls the strength of the KL divergence penalty. The D-optimal criterion seeks to optimize the following objective,

$$S_n^* = \arg \max_{S \subset \mathcal{D}, \, |S| = n} \log \det H(S), \tag{12}$$

which maximizes the volume of the confidence ellipsoid for the policy parameters and promotes better convergence of DPO. Problem equation 12 is however NP-hard (Welch, 1982; Allen-Zhu et al., 2021), as it is a combinatorial optimization over $\binom{|\mathcal{D}|}{n}$ subsets. To overcome this computational challenge, we further propose a greedy and iterative sample-selection strategy to incrementally optimize information gain (Nemhauser & Wolsey, 1978; Krause et al., 2008).

**Incremental Greedy Information Maximization.**; To overcome the combinatorial optimization problem, we exploit the matrix-determinant identity

$$\log \det (H + vv^\top) = \log \det H + \log (1 + v^\top H^{-1} v), \tag{13}$$

which is valid for any positive-definite matrix $H$ and vector $v$. We initialize the design matrix by $H_0 = \gamma I$, and the incremental greedy algorithm adds one negative at a time. At iteration $k$ it selects

$$i_k = \arg \max_{i \notin S_{k-1}} v_i^\top H_{k-1}^{-1} v_i, \quad S_k = S_{k-1} \cup \{i_k\}, \quad H_k = H_{k-1} + v_{i_k} v_{i_k}^\top, \tag{14}$$

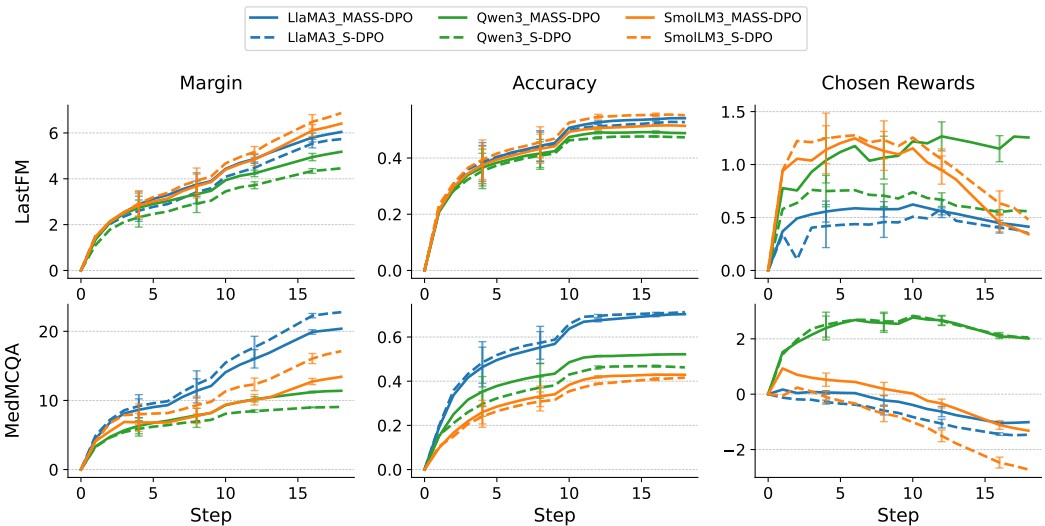

Figure 2: Margin, accuracy, and chosen reward comparisons on LastFM and MedMCQA datasets. MASS-DPO maintains consistently larger margins and higher accuracies, clearly illustrating the advantage of optimal negative selection over the random weighting approach used in SDPO.

and updates $H_k^{-1}$ via the Sherman-Morrison formula (Sherman & Morrison, 1950) in $\mathcal{O}(d^2)$ time. The term $v_i^\top H_{k-1}^{-1} v_i$ is the covariance matrix $H_{k-1}$ induced norm of $v_i$, so each step chooses the negative probing the *least explored* direction of the parameter space (Kveton et al., 2025). We illustrate this algorithm in detail in Algorithm 1. We formally demonstrate that this greedy Algorithm 1 achieves the same objective value as the intractable global optimum in Equation (12).

**Lemma 4.3** (Optimality of the Incremental Greedy Algorithm). *With $H_0 = \gamma I$ and $\gamma > 0$, the subset $S_n$ produced by the above procedure satisfies*

$$\log \det H(S_n) = \log \det H(S_n^*), \tag{15}$$

*where $S_n^\star$ is the maximizer in equation 12.*

We provide the proof of Lemma 4.3 in Appendix A.4. Lemma 4.3 ensures that the incremental selection mechanism embedded in MASS-DPO realizes the maximal Fisher information attainable with exactly $n$ negatives, justifying the finite-sample error bounds and convergence rates established in Section 5. Empirically, this property translates into faster alignment with a fraction of the computational cost required by exhaustive negative processing.

## 5 THEORETICAL ANALYSIS

We analyze the generalization performance of MASS-DPO under the linearized setting introduced above. Our goal is to quantify how the error in the estimated policy parameters affects the quality of logit predictions across all possible negative samples. To do so, we rely on a few standard assumptions necessary for our analysis (feature structure, boundedness, diversity, and culminate in a finite-sample generalization guarantee); the full statements are deferred to Appendix B. The analysis then proceeds to our main finite-sample guarantees.

**Theorem 5.1** (Maximum Logit Error Bound).

*Let $\hat{\theta}_n = \arg\min_{\theta \in \Theta} L(\theta; \mathcal{S}_n)$. Then the maximum logit error under is*

$$\mathcal{E}(\hat{\theta}_n, \theta_*) = \tilde{O}(d\sqrt{\log(1/\delta)/n})$$

*with probability at least $1 - \delta$, where $\tilde{O}$ hides all logarithmic factors but those in $\delta$.*

**Theorem 5.2** (Batch Design Estimation Error).

*With probability at least $1 - \delta$, given the total dataset $S_{k,n}$ of $k$ prompts and selected $n$ negative samples per prompt, the deviation of the estimated parameter from the true optimum is bounded in the $\Sigma_{k,n}$-norm:*

$$\left\| \hat{\theta}_{k,n} - \theta_* \right\|_{\Sigma_{k,n}} \leq \sqrt{\frac{d}{4} \log \left( \frac{1}{\delta} + \frac{k \cdot c_{\min}/\gamma}{(1 - c_{\min} \cdot k/\gamma)^{1/d} \cdot \delta} \right)} + 2\gamma^{1/2}. \tag{16}$$

*where $\gamma > 0$ is the regularization constant used to define $\Sigma_{k,n} = \gamma I + \nabla^2 L(\theta^*; S_{k,n})$, and the weighted sum of the features is $\phi = \sum_{i \in S_n} p_i \phi_i$.*

This follows (Abbasi-Yadkori et al., 2011; Kveton et al., 2025) by treating the S-DPO loss as a generalized linear model and applying self-normalized concentration bounds to the stochastic gradients. In practice, Theorem 5.1 suggests that with only a small number of negatives, MASS-DPO can already achieve bounded logit error, which translates into faster convergence in training. Theorem A.1 and Theorem 5.2 further imply that the selected negatives ensure stable generalization and better margin improvements across prompts, which we will verify in our experiments Section 6.

# 6 EXPERIMENTS

**Datasets.** Following recent DPO-based recommendation work (Chen et al., 2024), (Sun et al., 2024), and (He et al., 2025) we utilize two widely adopted real-world recommendation benchmarks: LastFM (Bertin-Mahieux et al., 2011) and MovieLens (Harper & Konstan, 2015). For QA tasks, we adopt two challenging multiple-choice QA datasets: MedMCQA (Pal et al., 2022), a medical-domain QA benchmark, and QASC (Khot et al., 2020), a scientific reasoning QA dataset. Following prior works (Rafailov et al., 2023), we report Accuracy, Margin, Chosen Rewards and several additional utility metrics, with detailed methodology available in Appendix B.2.

**Methods.** We benchmark MASS-DPO against established preference alignment approaches, categorized into pairwise methods, DPO (Rafailov et al., 2023) and its multi-negative extension DPO-k, and multi-negative methods, Softmax-DPO (SDPO) (Chen et al., 2024) and DMPO (Bai et al., 2024). To maintain fairness and manage computational costs, the number of negative candidates during training is set to 3 for all multi-negative methods (DPO-k, DMPO, SDPO, MASS-DPO) and 1 for DPO. However, for evaluation and test sets, we include all available negative candidates to better assess the model's ability to select the best sample from a larger pool (e.g., 20 candidates), thereby increasing the search space and providing a more robust measure of real-world performance.

Implementation details are provided in Appendix B.3. Our experiments are designed to validate the theoretical insights in Sections 4 and 5. In particular:

**LLM Usage:** We used large language models solely for grammar refinement and minor wording edits in drafting parts of this paper.

## 6.1 HOW WELL DOES MASS-DPO OPTIMIZE THE MULTI-NEGATIVE PREFERENCE LEARNING OBJECTIVE?

We compare MASS-DPO's active negative selection to the softmax-based random selection in SDPO on recommendation (LastFM) and QA (MedMCQA). Figure 2 tracks three alignment metrics during training: *margin* (logit gap between preferred vs. rejected), *accuracy*, and *chosen rewards*. Additional results for Movielens and QASC appear in Appendix B.2. Across experiments, MASS-DPO (solid) achieves larger margins and faster early gains than SDPO (dashed) on both datasets, with the gap emerging early and persisting through training. *Accuracy* follows the same pattern: curves for MASS-DPO rise more quickly and attain consistently higher plateaus. Finally, *chosen-reward trajectories* under MASS-DPO are smoother and more stable across steps, while SDPO exhibits noticeably noisier dynamics. Taken together, these trends indicate that actively selecting informative negatives leads to more efficient optimization of the multi-negative preference objective than random softmax selection.

| Model | Setting | Medmcqa | QASC | LastFM | MovieLens | Avg↑ |
|-------|---------|---------|------|--------|-----------|------|
| | DPO | 43.49 | 68.43 | 45.75 | 31.96 | 47.41 |
| | DMPO | 28.91 | 66.78 | 43.40 | 25.66 | 41.19 |
| Qwen3 | DPO-k | 55.56 | _71.96_ | _51.10_ | 44.56 | _55.80_ |
| | S-DPO | _52.56_ | 71.08 | 50.25 | **48.19** | 55.52 |
| | MASS-DPO | **56.66** | **72.19** | **52.30** | _47.58_ | **57.18** |
| | DPO | 33.27 | 67.00 | 51.90 | 37.60 | 47.44 |
| | DMPO | 25.50 | 65.23 | 50.10 | 28.68 | 42.38 |
| SmolLM3 | DPO-k | 44.09 | _69.98_ | 55.70 | 51.36 | 55.28 |
| | S-DPO | **44.99** | 69.43 | _55.90_ | **55.70** | _56.50_ |
| | MASS-DPO | _44.19_ | **71.63** | **57.25** | _54.03_ | **56.78** |
| | DPO | 52.25 | 71.08 | 54.60 | 33.52 | 52.86 |
| | DMPO | 25.70 | 69.87 | 49.95 | 28.18 | 43.42 |
| Llama3 | DPO-k | 71.04 | _73.95_ | 55.65 | 44.46 | 61.27 |
| | S-DPO | **72.19** | **74.61** | _56.55_ | _49.55_ | **63.23** |
| | MASS-DPO | _71.29_ | 73.62 | **57.35** | **49.70** | 62.99 |

Table 1: Accuracy (%) on four tasks across three base models. **Bold** = best, underlined = second best.

| Model | Method | MedMCQA MRR/Margin | QASC MRR/Margin | LastFM MRR/Margin | MovieLens MRR/Margin | Average↑ MRR/Margin |
|-------|--------|--------------------|-----------------|-------------------|----------------------|---------------------|
| Qwen3 | S-DPO | 69.74 / 9.33 | 81.87 / 5.29 | 64.12 / 4.67 | 61.29 / **4.98** | 69.26 / 6.07 |
| | MASS-DPO | **73.30 / 11.76** | **82.71 / 6.22** | **66.28 / 5.51** | **61.61** / 4.94 | **70.97 / 7.11** |
| SmolLM3 | S-DPO | **66.64 / 19.10** | 81.64 / **8.07** | 70.13 / **7.62** | **68.73** / 7.32 | 71.78 / 10.53 |
| | MASS-DPO | 66.30 / 14.42 | **82.73** / 7.91 | **70.79** / 7.29 | 68.13 / **7.41** | **71.99** / 9.26 |
| Llama3 | S-DPO | **83.44 / 23.93** | **84.13** / 7.26 | 70.58 / 6.42 | 63.92 / 5.20 | 75.52 / **10.70** |
| | MASS-DPO | 82.86 / 21.36 | 84.06 / 7.24 | **70.71 / 6.53** | **65.58 / 5.75** | **75.80** / 10.22 |

Table 2: MRR and Margin across four datasets. Each cell shows *MRR / Margin*.

## 6.2 HOW DOES MASS-DPO IMPROVE DOWNSTREAM POLICY PERFORMANCE COMPARED TO EXISTING PREFERENCE OPTIMIZATION METHODS?

To evaluate MASS-DPO's effectiveness in downstream policy performance we benchmark its performance against established preference optimization methods: DPO, DMPO, DPO-k, and SDPO on four datasets (MedMCQA, QASC, LastFM, MovieLens) using *Accuracy*. Results are reported for three base models in Table 1. MASS-DPO consistently outperform baselines across datasets and language models. Notably, MASS-DPO outperforms prior methods on average for Qwen3 and SmolLM3, and is a close second to S-DPO for Llama3. Per-dataset, MASS-DPO wins the majority of tasks for Qwen3 and matches or surpasses the strongest baseline on two tasks for both SmolLM3 and Llama3, while remaining top–2 in every dataset.

By contrast, simpler pairwise methods (DPO, DMPO, DPO-k) consistently underperform across tasks, emphasizing the limitations of these methods in effectively exploiting multiple negative samples. Although SDPO integrates multiple negatives through a softmax-based weighting, its lower performance relative to MASS-DPO demonstrates the critical importance of strategic negative selection rather than random or heuristic selection. These empirical results directly confirm our theoretical insights, maximizing the D-optimal objective (Eq. 12) selects complementary negatives that expand Fisher-information coverage, yielding higher downstream accuracy than random/softmax weighting (S-DPO) and pairwise methods that do not strategically use multiple negatives.

## 6.3 HOW DOES MASS-DPO ACHIEVE BETTER NEGATIVE SELECTION?

We assess negative-selection quality using downstream utility metrics, MRR and Margin (Table 2), and ranking quality on recommendation and QA via Recall/NDCG at $k \in \{1, 3\}$ (Tables 3 and 5). Across base models and datasets, MASS-DPO consistently improves *MRR* over S-DPO, while delivering higher or comparable *Margins*. On ranking metrics, MASS-DPO attains the most or tied-best scores in a majority of {R@1, N@1} cells and remains competitive at {R@3, N@3}, indicating better performance on both recommendation and QA. These results demonstrate that active neg-

| Model | Method | LastFM | | | | MovieLens | | | |
|---|---|---|---|---|---|---|---|---|---|
| | | R@1 | R@3 | N@1 | N@3 | R@1 | R@3 | N@1 | N@3 |
| Qwen3 | DPO | 46.15 | 72.60 | 46.15 | 61.60 | 29.64 | 59.48 | 29.64 | 46.89 |
| | DMPO | 44.50 | 72.05 | 44.50 | 60.57 | 24.50 | 56.30 | 24.50 | 42.88 |
| | DPO-k | 49.50 | 76.45 | 49.50 | 65.36 | 41.63 | 68.95 | 41.63 | 57.71 |
| | S-DPO | 48.55 | 75.10 | 48.55 | 64.14 | 45.92 | 71.47 | 45.92 | 60.86 |
| | MASS-DPO | **51.10** | **77.20** | **51.10** | **66.48** | **45.97** | **71.52** | **45.97** | **61.10** |
| SmolLM3 | DPO | 51.70 | 78.15 | 51.70 | 67.29 | 37.25 | 65.68 | 37.25 | 53.77 |
| | DMPO | 50.30 | 77.90 | 50.30 | 66.54 | 28.23 | 60.43 | 28.23 | 47.02 |
| | DPO-k | 56.30 | 80.55 | 56.30 | 70.71 | 51.01 | 75.71 | 51.01 | 65.41 |
| | S-DPO | 55.60 | **81.35** | 55.60 | 70.84 | **55.09** | **78.18** | **55.09** | **68.64** |
| | MASS-DPO | **57.05** | 80.70 | **57.05** | **71.08** | 54.18 | 77.57 | 54.18 | 68.03 |
| Llama3 | DPO | 55.15 | 80.35 | 55.15 | 70.06 | 34.48 | 63.56 | 34.48 | 51.31 |
| | DMPO | 49.95 | 78.35 | 49.95 | 66.76 | 27.82 | 58.72 | 27.82 | 45.69 |
| | DPO-k | 56.05 | 80.30 | 56.05 | 70.41 | 43.95 | 70.77 | 43.95 | 59.69 |
| | S-DPO | 56.50 | 80.85 | 56.50 | 70.95 | 48.94 | 73.39 | 48.94 | 63.25 |
| | MASS-DPO | **56.60** | **81.15** | **56.60** | **71.17** | **50.66** | **76.01** | **50.66** | **65.57** |

Table 3: Recall (R) and NDCG (N) at k={1,3} on LastFM and MovieLens.

| (a) $\beta$ ablation | | | | | | (b) Negatives $k$ ablation | | | | | |
|---|---|---|---|---|---|---|---|---|---|---|---|
| Model | $\beta$ | Medmcqa | QASC | LastFM | MovieLens | Model | $k$ | Medmcqa | QASC | LastFM | MovieLens |
| Qwen3 | 0.1 | **56.66**$_{0.77}$ | **72.19**$_{1.05}$ | **52.30**$_{0.79}$ | **47.58**$_{0.80}$ | Qwen3 | 1 | 50.95$_{0.78}$ | 68.21$_{1.13}$ | 47.80$_{0.80}$ | 32.86$_{0.73}$ |
| | 0.5 | 46.29$_{0.79}$ | 71.41$_{1.06}$ | 48.15$_{0.81}$ | 39.82$_{0.79}$ | | 3 | 56.66$_{0.77}$ | 72.19$_{1.05}$ | 52.30$_{0.79}$ | 47.58$_{0.80}$ |
| | 1.0 | 43.49$_{0.77}$ | 69.65$_{1.06}$ | 44.15$_{0.80}$ | 34.12$_{0.74}$ | | 5 | **57.31**$_{0.75}$ | **73.73**$_{1.03}$ | **54.50**$_{0.79}$ | **58.11**$_{0.78}$ |
| SmolLM3 | 0.1 | **44.19**$_{0.79}$ | **71.63**$_{1.07}$ | **57.25**$_{0.79}$ | 54.03$_{0.77}$ | SmolLM3 | 1 | 29.26$_{0.72}$ | 65.67$_{1.09}$ | 50.70$_{0.81}$ | 34.58$_{0.75}$ |
| | 0.5 | 39.73$_{0.76}$ | **71.63**$_{1.03}$ | 54.75$_{0.79}$ | **56.30**$_{0.79}$ | | 3 | 44.19$_{0.79}$ | **71.63**$_{1.07}$ | 57.25$_{0.79}$ | 54.03$_{0.77}$ |
| | 1.0 | 35.42$_{0.75}$ | 68.98$_{1.06}$ | 52.00$_{0.80}$ | 52.12$_{0.80}$ | | 5 | **46.59**$_{0.79}$ | **71.63**$_{1.04}$ | **59.50**$_{0.77}$ | **65.07**$_{0.74}$ |
| Llama3 | 0.1 | **71.29**$_{0.74}$ | **73.62**$_{1.03}$ | **57.35**$_{0.81}$ | 49.70$_{0.80}$ | Llama3 | 1 | 46.99$_{0.78}$ | 71.96$_{1.03}$ | 52.70$_{0.81}$ | 32.71$_{0.73}$ |
| | 0.5 | 69.69$_{0.74}$ | 73.51$_{1.01}$ | 55.75$_{0.80}$ | **51.06**$_{0.78}$ | | 3 | 71.29$_{0.74}$ | 73.62$_{1.03}$ | 57.35$_{0.81}$ | 49.70$_{0.80}$ |
| | 1.0 | 66.28$_{0.76}$ | 72.19$_{1.02}$ | 52.25$_{0.81}$ | 45.92$_{0.78}$ | | 5 | **73.55**$_{0.71}$ | **74.94**$_{0.98}$ | **60.05**$_{0.80}$ | **60.99**$_{0.77}$ |

Table 4: MASS-DPO ablations on two hyperparameters. (a) Varying the scale $\beta$ (0.1, 0.5, 1.0) while holding the number of negatives $k$ fixed. (b) Varying $k$ (1, 3, 5) while holding $\beta$ fixed.

ative selection via MASS-DPO effectively highlights and corrects policy weaknesses, enhancing downstream alignment and utility beyond standard optimization methods.

## 6.4 ABLATION STUDIES

MASS-DPO's behavior is governed by the Fisher-information structure (Equation (11)) and the D-optimal selection objective (Equation (12)). We therefore ablate two key knobs predicted by theory to matter most: the KL regularization scale $\beta$ and the number of selected negatives $k$.

### 6.4.1 EFFECT OF $\beta$

The coefficient $\beta$ tunes the pull toward the reference model and, through the softmax weights $p_j$, scales each candidate's Fisher contribution $v_j = \sqrt{p_j}\,\tilde{\phi}_j$ in the D-optimal criterion. Larger $\beta$ sharpens $p_j$ and can better separate informative from redundant negatives, but too much regularization restricts useful policy updates. Sweeping $\beta \in \{0.1, 0.5, 1.0\}$ across three model families (Table 4a), we find $\beta = 0.1$ consistently yields the strongest results.

### 6.4.2 NUMBER OF NEGATIVES ($k$)

D-optimal design predicts that adding more negatives improves parameter estimation until coverage of the information space saturates. Varying $k \in \{1, 3, 5\}$ with our greedy selector (Table 4b) shows monotonic gains from $k=1 \to 3 \to 5$ across models and datasets. These results indicate the greedy procedure reliably assembles complementary negatives that expand $\log \det$ of the information matrix, aligning empirical improvements with our D-optimal design analysis.

## 7 CONCLUSION

In this work, we introduced MASS-DPO, a theoretically grounded approach to active negative sample selection for multi-negative direct preference optimization. By formulating negative sampling as a D-optimal design problem, we effectively addressed redundancy and computational inefficiencies inherent in existing methods. Our incremental greedy algorithm ensures computational feasibility while retaining theoretical optimality. Theoretical analyses confirm the efficiency and convergence guarantees of MASS-DPO, and comprehensive experiments illustrate its superior performance and scalability across diverse language modeling and both recommendation and QA tasks.

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

## A APPENDIX

**Lemma A.1** (Gradient Derivation). *Consider the loss for a single sample*

$$L(\theta) = -\log \sigma\Big(Z(\theta)\Big), \quad with \quad Z(\theta) = -\log\left(\sum_{j=1}^{n} \exp\Big[\beta\left(\phi_j^\top \theta + b_j\right)\Big]\right), \qquad (17)$$

$$\frac{d}{dz}\Big[-\log\sigma(Z(\theta))\Big] = -\frac{1}{\sigma(Z(\theta))} \cdot \sigma'(Z(\theta)) = -\frac{\sigma(Z(\theta))(1-\sigma(Z(\theta)))}{\sigma(Z(\theta))} = -(1-\sigma(Z(\theta))).$$

$$\frac{\partial L}{\partial Z(\theta)} = -(1-\sigma(Z(\theta))). \qquad (18)$$

$$A(\theta) = \sum_{j=1}^{n} \exp\Big[\beta\left(\phi_j^\top \theta + b_j\right)\Big],$$

*so that $Z(\theta) = -\log A(\theta)$. Then,*

$$\frac{\partial Z(\theta)}{\partial \theta} = -\frac{1}{A(\theta)}\frac{\partial A(\theta)}{\partial \theta},$$

$$\frac{\partial A(\theta)}{\partial \theta} = \sum_{j=1}^{n} \exp\Big[\beta\left(\phi_j^\top \theta + b_j\right)\Big]\beta\phi_j,$$

$$\frac{\partial Z(\theta)}{\partial \theta} = -\beta \sum_{j=1}^{n} \frac{\exp\left[\beta\left(\phi_j^\top \theta - b_j\right)\right]}{A(\theta)} \phi_j = -\beta \sum_{j=1}^{n} p_j \phi_j,$$

*where the softmax weights are defined as*

$$p_j = \frac{\exp\left[\beta\left(\phi_j^\top \theta - b_j\right)\right]}{A(\theta)}.$$

$$\frac{\partial L}{\partial \theta} = \frac{\partial L}{\partial Z(\theta)} \cdot \frac{\partial Z(\theta)}{\partial \theta} = -(1 - \sigma(Z(\theta))) \cdot \left[-\beta \sum_{j=1}^{n} p_j \phi_j\right] = \beta(1 - \sigma(Z(\theta))) \sum_{j=1}^{n} p_j \phi_j.$$

*Thus, the gradient of the loss is*

$$\nabla_\theta L = \beta(1 - \sigma(Z(\theta))) \sum_{j=1}^{n} p_j \phi_j. \tag{19}$$

**Lemma A.2** (Hessian Derivation). *Recall the multi-negative DPO loss:*

$$L(\theta; S_n) = -\log \sigma\left(-\log \sum_{i \in S_n} \exp(\beta(\phi_i^\top \theta + b_i))\right),$$

*where $\sigma(\cdot)$ denotes the sigmoid function, and we define the shorthand*

$$Z_n = -\log \sum_{i \in S_n} \exp(\beta(\phi_i^\top \theta + b_i)), \quad p_j = \frac{\exp(\beta(\phi_j^\top \theta + b_j))}{\sum_{k \in S_n} \exp(\beta(\phi_k^\top \theta + b_k))}, \quad \phi = \sum_{j \in S_n} p_j \phi_j.$$

*Starting from the gradient Equation (9),*

$$\nabla_\theta L(\theta; S_n) = \beta(1 - \sigma(Z_n)) \sum_{j \in S_n} p_j \phi_j,$$

*we derive the Hessian by differentiating again with respect to $\theta$:*

$$\nabla_\theta^2 L(\theta; S_n) = \beta \nabla_\theta \left[(1 - \sigma(Z_n)) \sum_{j \in S_n} p_j \phi_j\right] \tag{20}$$

$$= \beta(1 - \sigma(Z_n)) \nabla_\theta \sum_{j \in S_n} p_j \phi_j + \beta \sigma(Z_n)(1 - \sigma(Z_n)) \sum_{j \in S_n} p_j \phi_j \nabla_\theta Z_n^\top. \tag{21}$$

*Expanding the first term using the definition of $p_j$ gives:*

$$\nabla_\theta \sum_{j \in S_n} p_j \phi_j = \beta \sum_{j \in S_n} p_j \phi_j \phi_j^\top - \beta \left(\sum_{j \in S_n} p_j \phi_j\right)\left(\sum_{j \in S_n} p_j \phi_j\right)^\top \tag{22}$$

$$= \beta \sum_{j \in S_n} p_j (\phi_j - \phi)(\phi_j - \phi)^\top. \tag{23}$$

*Note also that:*

$$\nabla_\theta Z_n = \beta \sum_{j \in S_n} p_j \phi_j = \beta \phi.$$

*Thus, substituting back, the Hessian becomes:*

$$\nabla_\theta^2 L(\theta; S_n) = \beta^2 (1 - \sigma(Z_n)) \sum_{j \in S_n} p_j (\phi_j - \phi)(\phi_j - \phi)^\top + \beta^2 \sigma(Z_n)(1 - \sigma(Z_n)) \phi \phi^\top \tag{24}$$

$$= \beta^2 (1 - \sigma(Z_n)) \left[\sigma(Z_n) \phi \phi^\top + \sum_{j \in S_n} p_j (\phi_j - \phi)(\phi_j - \phi)^\top\right]. \tag{25}$$

**Remark 1** (Loewner lower bound in Eq. Equation (10)). *From the decomposition above,*

$$\nabla_\theta^2 L(\theta; S_n) = \beta^2 (1 - \sigma(Z_n)) \Big[ \sigma(Z_n)\, \phi\phi^\top + \sum_{j \in S_n} p_j (\phi_j - \phi)(\phi_j - \phi)^\top \Big].$$

*The rank one term $\sigma(Z_n)\, \phi\phi^\top$ is positive semidefinite, and the weighted covariance term is also positive semidefinite. Therefore dropping the rank one term yields the Loewner lower bound*

$$\nabla_\theta^2 L(\theta; S_n) \succeq \beta^2 (1 - \sigma(Z_n)) \sum_{j \in S_n} p_j (\phi_j - \phi)(\phi_j - \phi)^\top,$$

*which is Equation (10).*

*This demonstrates how the Hessian measures curvature based on the variance of feature differences under the softmax weights $p_j$, capturing essential geometric insights into policy optimization.*

**Lemma A.3** (PSD lower bound used in Eq. (10)). *Under Assumption 4.1 and the definitions in Lemma 4.2, the Hessian of $L(\theta; S_n)$ satisfies*

$$\nabla_\theta^2 L(\theta; S_n) \succeq \beta^2 \big(1 - \sigma(Z_n)\big) \sum_{j=1}^n p_j (\phi_j - \phi)(\phi_j - \phi)^\top.$$

*Proof.* From Lemma 4.2,

$$\nabla_\theta^2 L(\theta; S_n) = \beta^2 \big(1 - \sigma(Z_n)\big) \Big[ \sigma(Z_n)\, \phi\phi^\top + \sum_{j=1}^n p_j (\phi_j - \phi)(\phi_j - \phi)^\top \Big].$$

Let $A := \sigma(Z_n)\, \phi\phi^\top$ and $B := \sum_{j=1}^n p_j (\phi_j - \phi)(\phi_j - \phi)^\top$. Because $\sigma(Z_n) \in (0,1)$, $A$ is a nonnegative scalar times an outer product, hence $A \succeq 0$. Each $(\phi_j - \phi)(\phi_j - \phi)^\top \succeq 0$ and $p_j \geq 0$, so $B \succeq 0$. Therefore $A + B \succeq B$. Multiplying by the nonnegative scalar $\beta^2 \big(1 - \sigma(Z_n)\big)$ preserves semidefinite order, giving the claimed bound. $\qquad\square$

**Theorem A.1** (Single Design Estimation Error). *Following (Kveton et al., 2025), to bound $\left\| \hat\theta_n - \theta_* \right\|_{\Sigma_n}$, we also show that $\|\nabla L(\theta_*; \mathcal{S}_n)\|_{\Sigma_n^{-1}}$ is small with high probability. We recall the from Lemma A.1,*

$$\nabla_\theta L(\theta_*; S_n) = -\beta \left(1 - \sigma(Z_n(\theta_*))\right) \sum_{j \in S_n} p_j \phi_j,$$

*where $Z_n(\theta_*) = \sum_{k \in S_n} \exp \left[ \beta(\phi_k^\top \theta_* - b_k) \right]$. Since the covariance matrix is lower-bounded as*

$$\Sigma_n \succeq V_n = c_{\min} \left( \frac{\gamma}{c_{\min}} I_d - (1 - \sigma(Z_n))\phi\phi^\top + \sum_{j \in \mathcal{S}_n} p_j \phi_j \phi_j^\top \right).$$

*Then we have*

$$\|\nabla L(\theta_*; \mathcal{S}_n)\|_{\Sigma_n^{-1}} \leq \frac{\beta(1 - \sigma(Z_n))}{\sqrt{c_{\min}}} \left\| \sum_{j \in \mathcal{S}_n} p_j \phi_j \right\|_{V_n^{-1}}.$$

*Following (Kveton et al., 2025), we have*

$$\left\| \sum_{i \in \mathcal{S}_n} p_i \phi_i \right\|_{V_n^{-1}} \leq \sqrt{ \frac{d}{4} \log \left( \frac{1 + (1 - c_{\min}(1 - \sigma(Z_n))\|\phi\|^2/\gamma)^{-1/d} \cdot \sum_{j \in \mathcal{S}_n} p_j \cdot c_{\min}\|\phi_j\|^2/\gamma}{\delta} \right) }$$

$$\leq \sqrt{ \frac{d}{4} \log \left( \frac{1 + (1 - c_{\min}\|\phi\|^2/\gamma)^{-1/d} \cdot c_{\min}/\gamma}{\delta} \right) }$$

*Then, with all equations, we have*

$$\left\|\hat{\theta}_n - \theta_*\right\|_{\Sigma_n} \leq \|\nabla L(\theta_*; \mathcal{S}_n)\|_{\Sigma_n^{-1}} + 2\gamma^{\frac{1}{2}}$$

$$\leq \sqrt{\frac{\beta^2 d}{c_{\min}} \log\left(\frac{1 + (1 - c_{\min}\|\phi\|^2/\gamma)^{-1/d} \cdot c_{\min}/\gamma}{\delta}\right)} + 2\gamma^{\frac{1}{2}}, \qquad (26)$$

*holds with probability at least $1 - \delta$.*

**Theorem A.2** (Batch Design Estimation Error). *Now we consider learning with all collected samples $\mathcal{S}_{k,n}$, where each $i$-th prompt is corresponded with $n$ negative samples actively collected $\mathcal{S}_{i,n}$. Following Lemma A.1, to bound $\left\|\hat{\theta}_{k,n} - \theta_*\right\|_{\Sigma_{k,n}}$, we also show that $\|\nabla L(\theta_*; \mathcal{S}_{k,n})\|_{\Sigma_{k,n}^{-1}}$ is small with high probability. We recall the from Lemma A.1,*

$$\nabla_\theta L(\theta_*; S_{k,n}) = -\beta \sum_{i=1}^{k} (1 - \sigma(Z_i(\theta_*))) \sum_{j \in S_{i,n}} p_{i,j}\phi_{i,j},$$

*where $Z_i(\theta_*) = \sum_{j \in S_{i,n}} \exp\left[\beta(\phi_{i,j}^\top \theta_* - b_{i,j})\right]$. Since the covariance matrix is lower-bounded as*

$$\Sigma_{k,n} \succeq V_{k,n} = c_{\min}\left(\frac{\gamma}{c_{\min}}I_d - \sum_{i=1}^{k}(1 - \sigma(Z_{i,n}))\phi_i\phi_i^\top + \sum_{i=1}^{k}\sum_{j \in \mathcal{S}_{i,n}} p_{i,j}\phi_{i,j}\phi_{i,j}^\top\right).$$

*Then we have*

$$\|\nabla L(\theta_*; \mathcal{S}_{k,n})\|_{\Sigma_{k,n}^{-1}} \leq \frac{\beta \sum_{i=1}^{k}(1 - \sigma(Z_{i,n}))}{\sqrt{c_{\min}}} \left\|\sum_{i=1}^{k}\sum_{j \in \mathcal{S}_{i,n}} p_{i,j}\phi_{i,j}\phi_{i,j}^\top\right\|_{V_{k,n}^{-1}}.$$

*Following (Kveton et al., 2025), we have*

$$\left\|\sum_{i \in \mathcal{S}_{k,n}} p_i\phi_i\right\|_{V_{k,n}^{-1}} \leq \sqrt{\frac{d}{4} \log\left(1/\delta + \frac{\sum_{i=1}^{k}\sum_{j \in \mathcal{S}_{i,n}} p_{i,j} \cdot c_{\min}\|\phi_{i,j}\|^2/\gamma}{\left(1 - c_{\min}\sum_{i=1}^{k}(1 - \sigma(Z_{i,n}))\|\phi\|^2/\gamma\right)^{1/d} \cdot \delta}\right)}$$

$$\leq \sqrt{\frac{d}{4} \log\left(1/\delta + \frac{k \cdot c_{\min}/\gamma}{(1 - c_{\min} \cdot k/\gamma)^{1/d} \cdot \delta}\right)}$$

*Then, with all equations, we have*

$$\left\|\hat{\theta}_{k,n} - \theta_*\right\|_{\Sigma_{k,n}} \leq \|\nabla L(\theta_*; \mathcal{S}_{k,n})\|_{\Sigma_{k,n}^{-1}} + 2\gamma^{\frac{1}{2}}$$

$$\leq \sqrt{\frac{d}{4} \log\left(1/\delta + \frac{k \cdot c_{\min}/\gamma}{(1 - c_{\min} \cdot k/\gamma)^{1/d} \cdot \delta}\right)} + 2\gamma^{\frac{1}{2}}, \qquad (27)$$

*holds with probability at least $1 - \delta$.*

**Lemma A.4** (Optimality of the Incremental Greedy Algorithm). *Let $(i_1, \dots, i_n)$ be the greedy indices and $H_k = \gamma I + \sum_{t=1}^{k} v_{i_t}v_{i_t}^\top$. Iterating equation 13 yields*

$$\det H_n = \det(\gamma I) \prod_{k=1}^{n}\left(1 + v_{i_k}^\top H_{k-1}^{-1}v_{i_k}\right). \qquad (28)$$

*Now consider any other subset $S = \{j_1, \dots, j_n\}$ (arbitrary order) and define $\tilde{H}_k$ analogously. Because $i_k$ maximises $v^\top H_{k-1}^{-1}v$ among the remaining candidates and $H_{k-1} \succeq \tilde{H}_{k-1}$, we have $v_{i_k}^\top H_{k-1}^{-1}v_{i_k} \geq v_{j_k}^\top \tilde{H}_{k-1}^{-1}v_{j_k}$ for every $k$. Applying equation 28 to both sequences and multiplying the $n$ inequalities delivers $\det H_n \geq \det \tilde{H}_n$, and hence $\det H(S_n) \geq \det H(S)$ for all admissible $S$. Taking logarithms completes the argument.*

---

**Algorithm 1** Greedy D-Optimal Multi-negative Active Sample Selection

---

1: **Input:** context $x$, preferred response $y^*$, candidate set $\mathcal{D} = \{y_i\}_{i=1}^N$, policy parameter $\theta$, scale $\beta$, number of negatives $n$
2: Compute feature differences and offsets, for each $i \in [N]$,
   $$\phi_i \leftarrow \phi(x, y_i) - \phi(x, y^*), \quad b_i \leftarrow \log \pi_{\text{ref}}(y^* \mid x) - \log \pi_{\text{ref}}(y_i \mid x)$$
3: Compute scores and softmax weights, for each $i \in [N]$,
   $$s_i \leftarrow \beta\big(\phi_i^\top \theta + b_i\big), \quad p_i \leftarrow \exp(s_i) / \sum_{k=1}^N \exp(s_k)$$
4: Center and weight features, for all $i \in [N]$
   $$\phi \leftarrow \sum_{j=1}^N p_j\, \phi_j, \quad \tilde{\phi}_i \leftarrow \phi_i - \phi, \quad v_i \leftarrow \sqrt{p_i}\, \tilde{\phi}_i$$
5: Initialize matrices and set: $H_0 \leftarrow \gamma \mathbf{I}_{d \times d}, S_0 \leftarrow \emptyset$
6: **for** $k = 1, \ldots, n$ **do**
7:    Incremental selection $I_k \leftarrow \arg\max_{i \in [N] \setminus S_{k-1}} \log\det\big(H_{k-1} + v_i v_i^\top\big)$
8:    Incremental update selection and design:    $S_k \leftarrow S_{k-1} \cup \{I_k\}, \quad H_k \leftarrow H_{k-1} + v_{I_k} v_{I_k}^\top$
9: **end for**
10: **Output:** selected negatives set $S_n$

---

## B  TECHNICAL ASSUMPTIONS

### B.1  ASSUMPTION DETAILS

**Assumption B.1** (Bounded Feature and Bias). *For any $(x, y)$ pair, the feature vectors and reference policy log-ratio are bounded:*
$$\|\phi(x, y)\|_2 \leq 1, \quad |b_i| \leq 1.$$
*Additionally, we constrain the parameter space to a unit ball: $\|\theta\|_2 \leq 1$.*

**Assumption B.2** (Bounded Design Weights). *Let $p_i = \exp[\beta(\phi_i^\top \theta - b_i)] / \sum_j \exp[\beta(\phi_j^\top \theta - b_j)]$. Then there exist constants $0 < c_{\min} \leq c_{\max} \leq \frac{1}{4}\beta^2$ such that:*

$$c_{\min} \leq \beta^2 (1 - \sigma(Z(\theta))) p_i \leq c_{\max}, \quad \forall i.$$

### B.2  EXPERIMENTAL SETTINGS

To further manage computational costs, we cap the number of response candidates at 20 for the LastFM, MovieLens, and MedMCQA datasets, and at 8 for QASC. Although MedMCQA natively provides only four options per question, we expand this to 20 by pooling all candidates with the same `subject_name` field. We also subsample each dataset to 20k training samples, 200 samples for online evaluation, and 2,000 samples for testing. All prompts are formatted using each model's provided chat template to ensure consistent input structure across tasks.

### B.3  IMPLEMENTATION DETAILS

We implement our experiments using PyTorch, leveraging three widely used pre-trained LLMs: *LlaMA-3.2-3B-Instruct* (Grattafiori et al., 2024), *SmolLM3* (Bakouch et al., 2025), and *Qwen3-4B* (Team, 2025). Each model is fine-tuned on 8 NVIDIA A100 GPUs with a per-device batch size of 2, gradient accumulation steps of 8, learning rate of $10^{-5}$, a cosine learning-rate scheduler with warmup ratio 0.05, and the Paged AdamW optimizer for 3 epochs with a fixed KL penalty coefficient at $\beta = 0.1$ across all experiments. More details included in We enable gradient check-pointing, gradient clipping is applied with a maximum norm of 0.3, and evaluation uses a batch size of 2. We extract representation vectors by mean-pooling the final hidden states, using either (a) all tokens from the concatenated prompt–response sequence or (b) only the response tokens, where prompt positions are masked out. Both strategies use the same pretrained LLM and tokenization pipeline.

**Hyperparameters.** For the D-optimal selection objective (Equation (11)), we set the ridge $\gamma = 0.1$ for all runs to ensure $H(S)$ is well conditioned, and we use the same $\beta$ as in training. In the main experiments $\beta = 0.1$, and we only vary $\beta$ in that ablation.

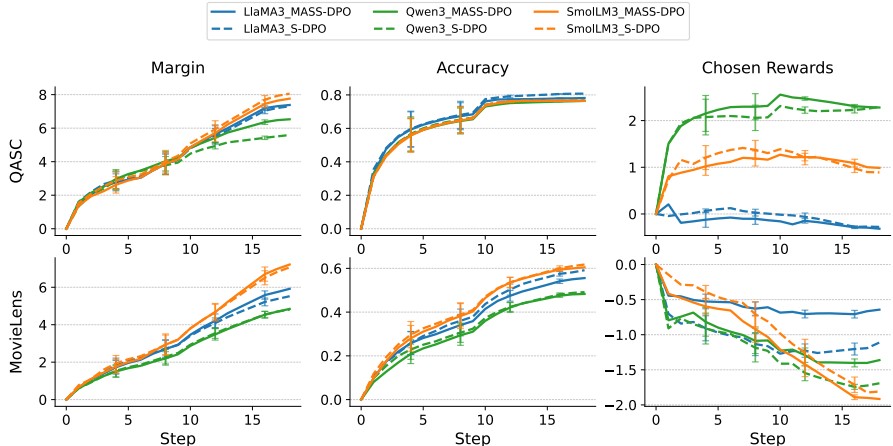

Figure 3: Comparison of MASS-DPO and SDPO on the MovieLens and QASC datasets. MASS-DPO consistently achieves higher margins, superior accuracy, and stable improvements in chosen rewards, highlighting the benefits of active negative sample selection. The x-axis (Step) counts the on-the-fly evaluations during training.

| Model | Method | MedMCQA | | | | QASC | | | |
|---|---|---|---|---|---|---|---|---|---|
| | | R@1 | R@3 | N@1 | N@3 | R@1 | R@3 | N@1 | N@3 |
| Qwen3 | DPO | 39.25 | 84.51 | 39.25 | 65.37 | 67.77 | 90.62 | 67.77 | 81.29 |
| | DMPO | 26.02 | 74.89 | 26.02 | 53.60 | 68.21 | 90.51 | 68.21 | 81.40 |
| | DPO-k | 54.59 | 89.67 | 54.59 | 74.86 | 71.08 | **92.72** | 71.08 | **83.95** |
| | S-DPO | 51.03 | 86.52 | 51.03 | 71.54 | 70.42 | 91.50 | 70.42 | 83.04 |
| | MASS-DPO | **56.34** | **89.72** | **56.34** | **75.62** | **71.85** | 91.61 | **71.85** | 83.73 |
| SmolLM3 | DPO | 33.33 | 81.75 | 33.33 | 61.01 | 67.11 | 90.07 | 67.11 | 80.70 |
| | DMPO | 26.02 | 75.39 | 26.02 | 53.93 | 66.11 | 88.74 | 66.11 | 79.54 |
| | DPO-k | 44.16 | 85.91 | 44.16 | 68.09 | 70.31 | 90.18 | 70.31 | 82.06 |
| | S-DPO | **45.46** | 87.22 | **45.46** | **69.54** | 70.53 | 91.39 | 70.53 | 82.80 |
| | MASS-DPO | 44.81 | **87.47** | 44.81 | 69.40 | **72.52** | **91.83** | **72.52** | **83.82** |
| Llama3 | DPO | 51.48 | 87.82 | 51.48 | 72.30 | 71.30 | 91.72 | 71.30 | 83.41 |
| | DMPO | 24.36 | 74.99 | 24.36 | 52.84 | 69.32 | 91.94 | 69.32 | 82.67 |
| | DPO-k | 71.13 | 93.88 | 71.13 | 84.51 | 73.95 | 92.38 | 73.95 | 84.93 |
| | S-DPO | **72.33** | 94.34 | **72.33** | **85.20** | **74.17** | 92.38 | **74.17** | **85.13** |
| | MASS-DPO | 71.23 | **94.49** | 71.23 | 84.84 | 73.84 | **92.60** | 73.84 | **85.13** |

Table 5: Recall (R) and NDCG (N) at k={1,3} on MedMCQA and QASC.

## B.4 RESULTS

