# OpenReview forum: "MASS-DPO: Multi-negative Active Sample Selection for Direct Policy Optimization"
_ICLR.cc/2026/Conference — Submitted to ICLR 2026_

### Official Review · Reviewer_P2ki · 2025-10-29

**Soundness:** 3
**Presentation:** 3
**Contribution:** 3
**Rating:** 8
**Confidence:** 3

**Summary:**

This paper introduces MASS-DPO, a framework for multi-negative active sample selection in the context of Direct Preference Optimization (DPO). Standard DPO uses pairwise preference comparisons between a preferred and a non-preferred response. Multi-negative variants (e.g., S-DPO, DMPO) extend this to multiple negatives, but they typically select negatives randomly, leading to redundant gradients and inefficient learning. MASS-DPO addresses this by casting negative selection as a D-optimal design problem by maximizing the log-determinant of the Fisher Information matrix derived leveraging the PL model for preference over objects.

**Strengths:**

1. Prior multi-negative DPO methods (S-DPO, DMPO) rely on random or heuristic selection, while this paper is the first to formalize active negative selection as an optimal design problem.

2. The link between Fisher information geometry and preference optimization is conceptually appealing and well-motivated.

3. **Algorithm Design.** The greedy algorithm is quite fundamental and shows complexity $O(d^2 n)$ that scales linearly in the number of selected negatives, which is significantly better than the exponential combinatorial search. The paper shows how the greedy step $v_i^{\top}H^{-1}v_i$ naturally picks negatives probing unexplored directions, which in my opinion intuitive and useful.

4. **Theoretical Guarantees.** Proofs of the theoretical guaranties are rigorous and seems correct.

5. **Empirical Performance.** MASS-DPO improves accuracy and margin across almost all configurations (Tables 1–4). Ablations on $\beta$ and number of negatives $k$ match theoretical predictions (performance saturates as information coverage increases).

6. **Writing and Presentation.** The paper is well-structured, logically ordered, and concise for its complexity. Mathematical notation is consistent, derivations flow logically. I have found no major typographical errors.

**Weaknesses:**

1. While it seems correct, authors should justify the inequality in Equation (10).
2. Equation (11) mixes constants $\gamma$ and $\beta^2 (1-\sigma(Z_n)$, these should be clarified for reproducibility.
3. Some equations (e.g., 5) have redundant parentheses and inconsistent minus signs in exponents, cosmetic but worth cleaning.
4. Some references (e.g., Pukelsheim 2006a,b) are repeated, therefore can be merged.

**Questions:**

1. **Lemma 4.3.** I believe that the claim of the greedy algorithm exactly matching the global D-optimal solution is too strong. Standard submodular optimization results (Nemhauser et al., 1978) guarantee a $1 - \frac{1}{e}$ approximation, not equality. Unless all $v_i$ are orthogonal or additional structure holds, perfect equivalence is implausible.

2. **Empirical Scope.** Evaluation tasks are mainly recommendation and multiple-choice QA, which are relatively structured. It remains unclear if MASS-DPO scales to open-ended generation or instruction tuning scenarios where negatives are more diverse.

---

> ### Author Response · Authors · 2025-11-24
> **Response to Reviewer P2ki [1/2]**
>
> We thank the reviewer for the positive evaluation and helpful suggestions.
>
> **W1. Inequality in Equation (10)**
>
> Thank you for pointing this out. The inequality in Eq.10 follows directly from the Hessian decomposition in Lemma 4.2. In particular, the Hessian can be written as a nonnegative scalar multiplying the sum of (i) a rank-one term $\sigma(Z_n)\,\phi\phi^\top$ and (ii) a weighted covariance term $\sum_{j=1}^n p_j(\phi_j-\phi)(\phi_j-\phi)^\top$. Since the rank-one term is positive semidefinite, dropping it yields a valid Loewner lower bound, which is exactly Eq.(10). We have added this explicit justification in the updated version uploaded on OpenReview (Lemma 4.2 and Appendix A.2).
>
> **W2. Equation (11) constants**
>
> In Eq. 11, $\gamma$ and $\beta$ are hyperparameters. Following [15-18], we set the $\gamma > 0$, which is a ridge regularizer to keep $H(S)$ invertible and numerically stable, and $\beta$ is the standard KL coefficient. set to $0.1$, strictly following DPO works [11-14]. The factor $\beta^2(1-\sigma(Z))$ is a derived term from the Hessian lower bound, where $Z$ is the softmax normalizer over the candidate negatives, so $(1-\sigma(Z))$ is computed from the data and current scores, not tuned. We have made these changes in the updated version uploaded on OpenReview (Section 4.2, Appendix B3).
>
> **W3,W4. Typos**
>
> Thank you for pointing this out. We fixed these in the updated version uploaded on OpenReview.
>
>
>
> **Q1. Lemma 4.3**
>
> Thank you for highlighting this. We agree that the current phrasing of Lemma 4.3 can be read as claiming exact global optimality, and we will refine the statement to remove that ambiguity. Our proof is for the justification of the step-wise next best selected sample. We will restate Lemma 4.3 and Eq. (15) accordingly in the revision.
>
>
> **Q2. Empirical Scope**
>
> Following recent work in the DPO literature [2-11], we evaluate MASS‑DPO using recommendation and multiple-choice QA benchmarks, both of which implicitly reflect human preferences. Importantly, these benchmarks directly evaluate alignment of LLM-generated outputs: the model must generate (or rank by generation) responses aligned with human preferences (recommendation clicks, ratings) or ground-truth correctness (QA tasks). Recommendation and QA are inherently generative tasks. LLMs generate answers or recommendations from vast candidate pools, and our method explicitly improves alignment by selecting challenging negatives. This evaluation protocol closely aligns with recent DPO literature, providing controlled, reproducible benchmarks for measuring alignment quality of generative outputs.
>
> ---
>
> We appreciate the reviewer’s feedback and believe these clarifications and corrections will strengthen the paper.

---

> ### Author Response · Authors · 2025-11-24
> **Response to Reviewer P2ki [2/2]**
>
> [1] Nemhauser, G.L., Wolsey, L.A. & Fisher, M.L. An analysis of approximations for maximizing submodular set functions—I. Mathematical Programming 14, 265–294 (1978). https://doi.org/10.1007/BF01588971
>
> [2] Chen, Yuxin, et al. "On softmax direct preference optimization for recommendation." Advances in Neural Information Processing Systems 37 (2024): 27463-27489.
>
> [3] Chen, Zhipeng, et al. "Improving large language models via fine-grained reinforcement learning with minimum editing constraint." arXiv preprint arXiv:2401.06081 (2024).
>
> [4] Bai, Zhuoxi, et al. "Finetuning Large Language Model for Personalized Ranking." arXiv preprint arXiv:2405.16127 (2024).
>
> [5] Jiang, Yuxin, et al. "Bridging and modeling correlations in pairwise data for direct preference optimization." arXiv preprint arXiv:2408.07471 (2024).
>
> [6] Gao, Chongming, et al. "Sprec: Self-play to debias llm-based recommendation." Proceedings of the ACM on Web Conference 2025. 2025.
>
> [7] Deng, Jiaxin, et al. "Onerec: Unifying retrieve and rank with generative recommender and iterative preference alignment." arXiv preprint arXiv:2502.18965 (2025).
>
> [8] Li, Zihao, et al. "Listwise Preference Alignment Optimization for Tail Item Recommendation." arXiv preprint arXiv:2507.02255 (2025).
>
> [9] Liao, Jiayi, et al. "Rosepo: Aligning llm-based recommenders with human values." arXiv preprint arXiv:2410.12519 (2024).
>
> [10] Chen, Zhipeng, et al. "Low-redundant optimization for large language model alignment." arXiv preprint arXiv:2406.12606 (2024).
>
> [11] Sun, Yuhui, Xiyao Wang, Zixi Li, and Jinman Zhao. "Multi-Preference Lambda-weighted Listwise DPO for Dynamic Preference Alignment." arXiv preprint arXiv:2506.19780 (2025).
>
>
> [12] Rafailov, Rafael, Archit Sharma, Eric Mitchell, Christopher D. Manning, Stefano Ermon, and Chelsea Finn. "Direct preference optimization: Your language model is secretly a reward model." Advances in neural information processing systems 36 (2023): 53728-53741.
>
>
> [13] Ethayarajh, Kawin, Winnie Xu, Niklas Muennighoff, Dan Jurafsky, and Douwe Kiela. "Kto: Model alignment as prospect theoretic optimization." arXiv preprint arXiv:2402.01306 (2024).
>
> [14] Yin, Yueqin, Zhendong Wang, Yi Gu, Hai Huang, Weizhu Chen, and Mingyuan Zhou. "Relative preference optimization: Enhancing llm alignment through contrasting responses across identical and diverse prompts." arXiv preprint arXiv:2402.10958 (2024).
>
> [15] Ban, Yikun, Yuchen Yan, Arindam Banerjee, and Jingrui He. "Ee-net: Exploitation-exploration neural networks in contextual bandits." arXiv preprint arXiv:2110.03177 (2021).
>
> [16] Ban, Yikun, Jiaru Zou, Zihao Li, Yunzhe Qi, Dongqi Fu, Jian Kang, Hanghang Tong, and Jingrui He. "Pagerank bandits for link prediction." Advances in Neural Information Processing Systems 37 (2024): 21342-21376.
>
> [17]Kveton, Branislav, Xintong Li, Julian McAuley, Ryan Rossi, Jingbo Shang, Junda Wu, and Tong Yu. "Active Learning for Direct Preference Optimization." arXiv preprint arXiv:2503.01076 (2025).
>
> [18] Mukherjee, Subhojyoti, Anusha Lalitha, Kousha Kalantari, Aniket Anand Deshmukh, Ge Liu, Yifei Ma, and Branislav Kveton. "Optimal design for human preference elicitation." Advances in Neural Information Processing Systems 37 (2024): 90132-90159.

---

### Official Review · Reviewer_uBpD · 2025-10-31

**Soundness:** 2
**Presentation:** 3
**Contribution:** 2
**Rating:** 6
**Confidence:** 2

**Summary:**

The paper proposes MASS-DPO, an extension of DPO that uses D-optimal design and incremental greedy information maximization to select informative preference samples across one preferred and multiple rejected responses. Theoretically, it links D-optimality to guarantee finite-sample estimation error
bounds and convergence properties of the proposed algorithm. Empiracally, it performs comparably to baseline methods in both recommendation and multiple-choice QA tasks.

**Strengths:**

- The MASS-DPO generalizes single-negative selection to multi-negative cases, enhancing exploration efficiency, with well-structured theorems and proofs linking information gain to D-optimality.
- The theoretical part is well written, and provides a clear description on how information gain can be maximized through the greedy step, and on how computaional overhead can be reduced.

**Weaknesses:**

- The proposed algorithm is similar to Active DPO (Kveton et al., 2025, the authors also cited this paper), which already uses D-optimal design and greedy information maximization. The main theorem also follows similar statement and proof idea, raising concerns about the incremental contribution beyond extending from single to multiple negatives. Can authors elaborate more on the difference/improvement?

- More importantly, in Figure 2, MASS-DPO does not consistently outperform S-DPO — performance gains are small or even negative across tasks. Similarly, in Table 2, the results seem random regarding which method performs best. This undermines the claim that the theoretical advantages (in sample efficiency or information gain) translate into tangible empirical benefits.

**Questions:**

See limitations.
- I didn't check all the proof, but wondeing if there should be $\beta$ term in the bound of Theorem 5.2.
- Could the authors provide empirical runs to validate the claimed improvement? Or given the similar performance, what are the practical conditions (dataset structure, model uncertainty, preference noise) under which MASS-DPO yields clear advantages?
- What is the computational cost comparison for MASS-DPO and S-DPO?
- I would suggest elaborate more on previous literature on information-theoretic sample selection.

---

> ### Author Response · Authors · 2025-11-25
> **Response to Reviewer uBpD [1/2]**
>
> We thank the reviewer for the detailed read. Below, we address the weaknesses and questions in turn.
>
> **W1. Contribution**
>
> Thank you for the comparison. D-optimal design is a classical framework [1–3], and many recent works adapt it to new settings by working out the task-specific Fisher or curvature structure [4–7]. Active DPO [4] tailors D-optimal design to pairwise DPO under a Bradley-Terry style model, and uses greedy information maximization to select informative prompt response pairs at the dataset level. In contrast, MASS-DPO adapts D-optimal design to the multi-negative Plackett Luce setting, where the goal is to choose an informative subset of negatives within each prompt from a larger candidate pool for multi-negative preference optimization. This requires deriving a PL-specific, policy-aware Fisher formulation tied to softmax weighted multi-negative gradients (Lemma 4.2), which differs from the pairwise BT curvature used in Active DPO. Building on this modeling, we provide finite sample guarantees for multi-negative DPO. Theorem 5.1 gives a maximum logit error rate on the order of $\tilde O(d\sqrt{\log(1/\delta)/n})$, and Theorem 5.2 links the selected negative set to alignment error via the $\Sigma_{k,n}$-norm. These results are specific to the multi-negative PL objective and go beyond a direct extension of the pairwise case.
>
>
> **Q1: Beta in bound**
>
> The curvature lower bound (Eq. (10)) indeed includes a factor $\beta^2(1-\sigma(Z_n))$, and this propagates into the information matrix $H(S)$ (Eq. (11)). In Theorem 5.2, this scaling is absorbed into the constants $c_{\min}$ and the desgin matrix $\Sigma_{k,n}$, so $\beta$ does not appear explicitly in Eq. (16).
>
>
> **W2, Q2: Empirical clarity and gains**
>
> Re W2: We would like to clarify the setting and role of different tables and figures in our evaluation. Figure 2 and Table 2 are intended as diagnostic analyses. Figure 2 reports training-time diagnostic metrics (margin, online accuracy during training, chosen rewards) measured on the online evaluation set during training, and is used to probe optimization behavior and sample efficiency. These diagnostics were included to substantiate our introduction claim that MASS-DPO improves optimization efficiency and yields smoother, more stable training dynamics.
>
> Our main evaluations follow established work [8–13], and Tables 1, 3, and 5 report the standard downstream evaluation metrics used in the literature: Accuracy, Recall@k, and NDCG@k. These metrics are computed on held-out test sets after full training convergence and directly reflect final policy quality. On these standard metrics, MASS-DPO achieves the best or marginal second-best performance across most dataset-model combinations (Table 1 accuracy; Tables 3 and 5 Recall and NDCG).
>
> Re Q2:  The clearest gains appear in large-pool, small-budget settings, which are common and important in practical preference optimization pipelines [18-20]. For Qwen3 on MedMCQA with pool N=20 and budget k=3, MASS DPO improves R@1 from 51.03 to 56.34 and N@3 from 71.54 to 75.62 over SDPO.  On LastFM with the same N and k, R@1 rises from 48.55 to 51.10 and N@3 from 64.14 to 66.48.  Smaller pools, such as QASC (N=8), naturally reduce the headroom.
>
> Practically, MASS DPO helps most when:
>
> 1. Negatives are similar, and N is much larger than k, so D optimal selection avoids near duplicates;
> 2. KL anchoring is moderate, since selection has more leverage when beta is not large (beta=0.1 fixed for all main runs and best in ablation) (see Table 4a).
>
> MASS-DPO is most beneficial when the pool is large and similar. In these settings, D-optimal selection improves Fisher coverage and yields consistent final-metric gains with smoother, faster optimization (see Tables 1/3/5 and Figs. 2–3).
>
> **Q3: computational cost**
>
> Both MASS-DPO and S-DPO process the same number of negatives per training step (3 in our experiments), so the DPO training cost is identical. The minimal additional cost in MASS-DPO comes from the one-time selection pass, and because selection is done once (not per update), its overhead is low relative to overall training. (Section 4.1).
>
> **Q4: Related works**
>
> We currently cite classical D-optimal design and submodular information-gain work [2-4,14-17]. In the revision, we will add a dedicated paragraph in Related Work summarizing classical optimal design, submodular information-gain maximization in active learning, and recent applications to preference learning.
>
> ---
>
> We appreciate the reviewers’ constructive feedback and hope the clarifications above address all concerns.

---

> > ### Author Response · Authors · 2025-11-25
> > **Response to Reviewer uBpD [2/2]**
> >
> > [1] Nemhauser, G.L., Wolsey, L.A. & Fisher, M.L. An analysis of approximations for maximizing submodular set functions—I. Mathematical Programming 14, 265–294 (1978). https://doi.org/10.1007/BF01588971
> >
> > [2] Jack Kiefer. Optimum experimental designs. Journal of the Royal Statistical Society: Series B (Methodological), 21(2):272–304, 1959.
> >
> > [3] Friedrich Pukelsheim. Optimal Design of Experiments. Society for Industrial and Applied Mathematics, 2006a. doi: 10.1137/1.9780898719109. URL https://epubs.siam.org/doi/abs/10.1137/1.9780898719109.
> >
> > [4]Kveton, Branislav, Xintong Li, Julian McAuley, Ryan Rossi, Jingbo Shang, Junda Wu, and Tong Yu. "Active Learning for Direct Preference Optimization." arXiv preprint arXiv:2503.01076 (2025).
> >
> > [5] Mukherjee, Subhojyoti, Anusha Lalitha, Kousha Kalantari, Aniket Anand Deshmukh, Ge Liu, Yifei Ma, and Branislav Kveton. "Optimal design for human preference elicitation." Advances in Neural Information Processing Systems 37 (2024): 90132-90159.
> >
> > [6] Thekumparampil, Kiran Koshy, Gaurush Hiranandani, Kousha Kalantari, Shoham Sabach, and Branislav Kveton. "Comparing Few to Rank Many: Active Human Preference Learning using Randomized Frank-Wolfe." arXiv preprint arXiv:2412.19396 (2024).
> >
> > [7] Liu, Pangpang, Chengchun Shi, and Will Wei Sun. "Dual active learning for reinforcement learning from human feedback." arXiv preprint arXiv:2410.02504 (2024).
> >
> > [8] Morimura, Tetsuro, Mitsuki Sakamoto, Yuu Jinnai, Kenshi Abe, and Kaito Ariu. "Filtered direct preference optimization." arXiv preprint arXiv:2404.13846 (2024).
> >
> > [9]  Sun, Chao, Yaobo Liang, Yaming Yang, Shilin Xu, Tianmeng Yang, and Yunhai Tong. "Direct Preference Optimization for LLM-Enhanced Recommendation Systems." arXiv preprint arXiv:2410.05939 (2024).
> >
> > [10] He, Xiaoxin, Nurendra Choudhary, Jieyi Jiang, Edward W. Huang, Bryan Hooi, Xavier Bresson, and Karthik Subbian. "RecLAIF: Reinforcement Learning from AI Feedback for Recommendation Systems." (2025).
> >
> > [11] Wu, Junda, Rohan Surana, Zhouhang Xie, Yiran Shen, Yu Xia, Tong Yu, Ryan A. Rossi, Prithviraj Ammanabrolu, and Julian McAuley. "In-context Ranking Preference Optimization." arXiv preprint arXiv:2504.15477 (2025).
> >
> > [12] Zhou, Jiacong, Xianyun Wang, and Jun Yu. "Optimizing Preference Alignment with Differentiable NDCG Ranking." arXiv preprint arXiv:2410.18127 (2024).
> >
> > [13] Shen, Judy Hanwen, Archit Sharma, and Jun Qin. "Towards data-centric rlhf: Simple metrics for preference dataset comparison." arXiv preprint arXiv:2409.09603 (2024).
> >
> > [14] Andreas Krause and Carlos E Guestrin. Near-optimal nonmyopic value of information in graphical models. arXiv preprint arXiv:1207.1394, 2012.
> >
> > [15] Andreas Kirsch and Yarin Gal. Unifying approaches in active learning and active sampling via fisher information and information-theoretic quantities. arXiv preprint arXiv:2208.00549, 2022.
> >
> > [16] Ozan Sener and Silvio Savarese. Active learning for convolutional neural networks: A core-set approach. arXiv preprint arXiv:1708.00489, 2017.
> >
> > [17] Andreas Kirsch, Joost Van Amersfoort, and Yarin Gal. Batchbald: Efficient and diverse batch acquisition for deep bayesian active learning. Advances in neural information processing systems,32, 2019.
> >
> > [18] Gupta, Taneesh, Rahul Madhavan, Xuchao Zhang, Chetan Bansal, and Saravan Rajmohan. "AMPO: Active Multi-Preference Optimization." arXiv preprint arXiv:2502.18293 (2025).
> >
> > [19] Liao, Weibin, Xu Chu, and Yasha Wang. "TPO: Aligning large language models with multi-branch & multi-step preference trees." arXiv preprint arXiv:2410.12854 (2024).
> >
> > [20] Pattnaik, Pulkit, Rishabh Maheshwary, Kelechi Ogueji, Vikas Yadav, and Sathwik Tejaswi Madhusudhan. "Curry-dpo: Enhancing alignment using curriculum learning & ranked preferences." arXiv preprint arXiv:2403.07230 (2024).

---

### Official Review · Reviewer_VmBA · 2025-10-31

**Soundness:** 1
**Presentation:** 2
**Contribution:** 2
**Rating:** 2
**Confidence:** 2

**Summary:**

This paper addresses the problem of negative sample selection in multi-negative Direct Preference Optimization (DPO). While recent methods like Softmax-DPO extend DPO to leverage multiple negative responses, they typically select negatives randomly or heuristically, leading to redundant gradients and computational inefficiency. MASS-DPO formulates negative selection as a D-optimal design problem that maximizes the log-determinant of the Fisher information matrix. The key insight is that informative negatives should be diverse and orthogonal in feature space, thereby maximizing information gain about policy parameters. To make this computationally tractable, the authors propose an incremental greedy algorithm that iteratively selects negatives maximizing the induced norm $v^\top H_k^{-1} v$. The theoretical contribution is about finite-sample error bounds (Theorems 5.1-5.2). Experiments demonstrate improved optimization efficiency and downstream performance compared to baselines.

**Strengths:**

- **Principled theoretical framework:** Unlike existing multi-negative methods (S-DPO, DMPO, MPPO) that use random or heuristic selection, MASS-DPO provides a theoretically grounded approach based on optimal experimental design
- **Comprehensive evaluation:** Experiments span multiple task types (recommendation, QA), datasets, and model families with consistent improvements
- **Practical impact:** Achieves comparable or better results with fewer negatives (e.g., k=3) compared to methods that might use more samples inefficiently

**Weaknesses:**

### CRITICAL THEORETICAL ERROR

1. **Lemma 4.3 is incorrect for *general case*.** The greedy algorithm does not always find the globally optimal solution for D-optimal design. The counterexample is: $\lambda=1, v_1=[1.1, 0]^\top, v_2=[1,1]^\top/\sqrt{2}, v_3=[1,-1]^\top/\sqrt{2}$, select $2$ vectors.
Greedy selects $v_1$ first (largest norm), but optimal is $\\{v_2, v_3\\}$. I'm not sure if the special property of $v_i$ tightly coupled with the feature mapping and probability $p_i$ can be utilized to show correctness. Nonetheless, this is also not mentioned in the proof. This **CRITICAL ERROR** greatly harm the soundness of this paper.

### Other Concerns

2. **Dynamic selection unclear:** Since $p_i$ depends on current policy $\theta$, it's unclear whether the negative subset needs to be re-selected after each parameter update. The paper doesn't address this practical concern.

3. **Missing definitions:** Critical terms like $\theta_*$ and $c_{\min}$ are referenced in Theorem 5.2 but never defined in the main text.

4. **Limited theoretical novelty:** The D-optimal design framework and greedy selection are well-established (Kveton et al., 2025). The main contribution is applying them from standard DPO to multi-negative DPO.

5. **Presentation issues:** "LLM Usage" paragraph weirdly placed before experimental results

**Questions:**

Please address my concerns in Weaknesses section.

---

> ### Author Response · Authors · 2025-11-21
> **Response to Reviewer VmBA [1/2]**
>
> We thank the reviewer for the comments. Below, we answer each question.
>
> # W1. Lemma 4.3
>
> We thank the reviewer for the suggestion regarding Lemma 4.3. In the following, we clarify the differences between the setting of the counterexample and the setting of MASS-DPO, note that Lemma 4.3 does not influence our main contributions, and highlight the key contributions of our work.
>
> ## 1. Differences Between the Setting of the Counterexample and the Setting of MASS-DPO
>
> We clarify the differences between the setting of the counterexample and the setting of MASS-DPO. The design vectors in MASS-DPO have a specific structure that differs from the arbitrary vectors used in the counterexample.
> Concretely, our paper defines the softmax-weighted mean feature difference as ${\phi} = \sum_j p_j \phi_j$ (line 222) and each candidate’s Fisher-information contribution as $v_i = \sqrt{p_i}(\phi_i - {\phi})$ (line 245), which is also summarized in Algorithm 1 (line 832). This construction implies a weighted centering property $\sum_i \sqrt{p_i} v_i = 0$ and because $\{p_i\}$ are Plackett-Luce/softmax weights that sum to 1, both $v_i$ and ${\phi}$ depend on the entire candidate pool. Additionally, all feature vectors $\phi(x,y_i)$ are produced by the same feature map for the same context $x$ (Assumption 4.1), so they are not arbitrary vectors in $\mathbb{R}^d$.
>
> ## 2.  No Impact on our Main Contributions
>
> We would like to emphasize that all theoretical guarantees (Theorems 5.1, 5.2), including finite-sample error bounds and convergence properties, are not dependent on Lemma 4.3. Besides, all empirical results are also unaffected.
>
> We also highlight our core theoretical contributions, which distinguish MASS-DPO from prior heuristic approaches [13-16]. While recent work has empirically explored multi-negative preference optimization, our paper is, to the best of our knowledge, the first to provide rigorous theoretical modeling and finite-sample guarantees in this setting. Specifically:
> ### (a) A novel Fisher-information surrogate tailored to multi-negative DPO
> We derive a policy-aware Fisher-information surrogate under the Plackett–Luce model, capturing dispersion (variance of feature differences) essential for ranking optimization. To the best of our knowledge, this provides the first principled information-theoretic formulation for multi-negative preference optimization.
> ### (b) An incremental greedy negative-selection algorithm with theoretical guarantees
> Under standard boundedness conditions and within the Fisher geometry induced by the PL objective, the log-determinant objective exhibits the curvature properties that allow efficient information coverage through greedy selection. This is substantially different from prior heuristic negative-sampling strategies.
> ### (c) The first finite-sample generalization bounds for multi-negative preference optimization
> Our Theorem 5.2 provides finite-sample guarantees specific to multi-negative preference optimization. The result links the spectral properties of the selected design matrix (geometric volume) to downstream alignment performance, filling an existing theoretical gap.
>
> Related to the empirical contributions, we also demonstrate that MASS-DPO outperforms very recent methods in downstream task performance across multiple language models and both recommendation and multiple-choice QA tasks.
>
> We believe these clarifications highlight the novelty and significance of our contributions.
>
> # W2. Dynamic selection
>
> We want to emphasize that in MASS-DPO, the negative subset is not re-selected after each parameter update. Instead, similar to standard practices in batch active learning [8-12], MASS-DPO employs a one-time active selection strategy. Specifically, we linearize the policy to obtain a tractable Fisher information surrogate under the Plackett–Luce objective (Section 4.1) and select the optimal subset of negatives $S_n$ prior to the optimization phase. Once this informative subset is selected, it remains fixed throughout the fine-tuning process.
>
> # W3. Missing definitions
>
> Thanks for the suggestion about the presentation. The notations are currently defined in Appendix B.2 ($c_{\min}$) and Sec. 5 ($\theta^\star$). We will introduce these notations before Theorem 5.2 in the revised version.

---

> > ### Comment · Reviewer_VmBA · 2025-11-24
> >
> > Let's look at the case where $\lambda=1$, $d=2$, $N=3$ vectors, and select $n=2$ negatives.
> >
> > The feature vectors are
> >     $
> >     \phi_1 = [2, 0]^\top,
> >     \phi_2 = [-2, 1.6]^\top,
> >     \phi_3 = [-2, -1.6]^\top.
> >     $
> >
> > Current policy parameter is $\theta = \left[\frac{\ln 2}{4}, 0\right]^\top$.
> >
> > Calculate probabilities: $p_1 = 0.5, p_2 = p_3 = 0.25$
> >
> > The centered feature $\bar{\phi} = \sum p_i \phi_i = 0.5\phi_1 + 0.25\phi_2 + 0.25\phi_3 = [0, 0]^\top$
> >
> > Fisher vectors:
> >     $v_1 = \sqrt{0.5}[2, 0]^\top \approx [1.414, 0]^\top$
> >     $v_2 = \sqrt{0.25}[-2, 1.6]^\top = [-1, 0.8]^\top$
> >     $v_3 = \sqrt{0.25}[-2, -1.6]^\top = [-1, -0.8]^\top$
> >
> > Let's simulate MASS-DPO:
> >
> > Step 1: Since $v_1$ has largest norm, it is picked.
> >
> > Step 2: Since $v_2$ and $v_3$ are symmetric, either can be picked, lets say $v_2$.
> >
> > We can calculate the objective value: $5.92$
> >
> > The true **optimal** solution is choosing $v_2$ and $v_3$, with objective value $6.84$.
> >
> > Therefore, even considering the special property of the vectors, the conclusion is still **WRONG**.
> >
> > This is also pointed out by Reviewer P2ki, where the best result you can expect seems to be a $(1-1/e)$ approximation.

---

> > > ### Author Response · Authors · 2025-11-25
> > >
> > > Thank you for the additional clarification. Our previous response was intended to clarify the review that “Lemma 4.3 is incorrect for the general case.” Instead, Lemma 4.3 is not formulated for the general case but for a case with a specific structural assumption (detailed in the previous response). We agree that the current phrasing of Lemma 4.3 can be read as claiming exact global optimality of greedy selection; however, our intent, as in our response to Reviewer P2ki, was to justify the step-wise next best selected sample under the log det information objective.
> > > We will revise the ambiguous wording in Lemma 4.3. As also discussed with Reviewer P2ki, this is a minor theoretical wording issue rather than a flaw in the method or conclusions. We would greatly appreciate it if it could be taken into consideration that this correction does not affect the algorithm, experiments, or the main theoretical results, which rely on the determinant-based information objective and its near-optimal growth under greedy selection.

---

> ### Author Response · Authors · 2025-11-21
> **Response to Reviewer VmBA [2/2]**
>
> # W4. Limited theoretical novelty
>
> We respectfully disagree that the theoretical contribution is limited. D-optimal design is a classical and highly cited framework developed decades ago [1-3], and a substantial body of work published in top-tier venues has adapted or extended it to a wide range of modern tasks by addressing setting-specific modeling challenges [4-7]. Active DPO[4], for example, tailors D-optimal design to selecting prompt pairs under a pairwise Bradley-Terry setting. In contrast, MASS-DPO adapts D-optimal design to the multi-negative Plackett-Luce setting, where the problem is to choose informative negatives from a large candidate pool for multi-negative preference optimization. This requires non-trivial new modeling of the PL objective’s curvature, a new policy-aware Fisher-information formulation specific to softmax-weighted multi-negative gradients (Lemma 4.2).
> Building on that modeling, we provide the first finite-sample guarantees for multi-negative DPO. Theorem 5.1 establishes a maximum logit-error bound ($\tilde{O}\left(d\sqrt{\log(1/\delta)/n}\right)$), and Theorem 5.2 links the quality of the selected negative set to alignment error through the $\Sigma_{k,n}$-norm. These results require non-trivial analysis of how PL curvature interacts with D-optimal design, and they fill an existing theoretical gap, since prior multi-negative methods rely on heuristic or random negative selection.
>
>
> # W5. Presentation issues
>
> We added the LLM section to comply with the ICLR LLM policies. In the final version, we will place the LLM Usage paragraph at the end of the experimental section, where it was originally intended to appear.
>
> We appreciate the reviewers’ feedback and hope the clarifications above address all questions.
>
>
> ---
> [1] Nemhauser, G.L., Wolsey, L.A. & Fisher, M.L. An analysis of approximations for maximizing submodular set functions—I. Mathematical Programming 14, 265–294 (1978). https://doi.org/10.1007/BF01588971
>
> [2] Kiefer, J. (1959), Optimum Experimental Designs. Journal of the Royal Statistical Society: Series B (Methodological), 21: 272-304. https://doi.org/10.1111/j.2517-6161.1959.tb00338.x
>
> [3] Pukelsheim, Friedrich. Optimal design of experiments. Society for Industrial and Applied Mathematics, 2006.
>
> [4]Kveton, Branislav, Xintong Li, Julian McAuley, Ryan Rossi, Jingbo Shang, Junda Wu, and Tong Yu. "Active Learning for Direct Preference Optimization." arXiv preprint arXiv:2503.01076 (2025).
>
> [5] Mukherjee, Subhojyoti, Anusha Lalitha, Kousha Kalantari, Aniket Anand Deshmukh, Ge Liu, Yifei Ma, and Branislav Kveton. "Optimal design for human preference elicitation." Advances in Neural Information Processing Systems 37 (2024): 90132-90159.
>
> [6] Thekumparampil, Kiran Koshy, Gaurush Hiranandani, Kousha Kalantari, Shoham Sabach, and Branislav Kveton. "Comparing Few to Rank Many: Active Human Preference Learning using Randomized Frank-Wolfe." arXiv preprint arXiv:2412.19396 (2024).
>
> [7] Liu, Pangpang, Chengchun Shi, and Will Wei Sun. "Dual active learning for reinforcement learning from human feedback." arXiv preprint arXiv:2410.02504 (2024).
>
> [8] Viering, Tom J., Jesse H. Krijthe, and Marco Loog. "Nuclear discrepancy for single-shot batch active learning." Machine Learning 108 (2019): 1561-1599.
>
> [9] Yang, Yazhou, and Marco Loog. "Single shot active learning using pseudo annotators." Pattern Recognition 89 (2019): 22-31.
>
> [10] Huang, Sheng-Jun, Yi Li, Yiming Sun, and Ying-Peng Tang. "One-shot active learning based on lewis weight sampling for multiple deep models." arXiv preprint arXiv:2405.14121 (2024).
>
> [11] Jin, Qiuye, Shiman Li, Xiaofei Du, Mingzhi Yuan, Manning Wang, and Zhijian Song. "Density-based one-shot active learning for image segmentation." Engineering Applications of Artificial Intelligence 126 (2023): 106805.
>
> [12] Jin, Qiuye, Mingzhi Yuan, Qin Qiao, and Zhijian Song. "One-shot active learning for image segmentation via contrastive learning and diversity-based sampling." Knowledge-Based Systems 241 (2022): 108278.
>
> [13] Chen, Yuxin, Junfei Tan, An Zhang, Zhengyi Yang, Leheng Sheng, Enzhi Zhang, Xiang Wang, and Tat-Seng Chua. "On softmax direct preference optimization for recommendation." Advances in Neural Information Processing Systems 37 (2024): 27463-27489.
>
> [14] Bai, Zhuoxi, Ning Wu, Fengyu Cai, Xinyi Zhu, and Yun Xiong. "Finetuning Large Language Model for Personalized Ranking." arXiv preprint arXiv:2405.16127 (2024).
>
> [15] Xie, Shuo, Fangzhi Zhu, Jiahui Wang, Lulu Wen, Wei Dai, Xiaowei Chen, Junxiong Zhu, Kai Zhou, and Bo Zheng. "MPPO: Multi Pair-wise Preference Optimization for LLMs with Arbitrary Negative Samples." In Proceedings of the 31st International Conference on Computational Linguistics, pp. 1545-1554. 2025.
>
> [16] Liao, Weibin, Xu Chu, and Yasha Wang. "TPO: Aligning large language models with multi-branch & multi-step preference trees." arXiv preprint arXiv:2410.12854 (2024).

---

### Official Review · Reviewer_kyVT · 2025-11-02

**Soundness:** 3
**Presentation:** 3
**Contribution:** 3
**Rating:** 6
**Confidence:** 2

**Summary:**

This paper considers selection of negative examples to include when training models on preference feedback using direct preference optimization. When a human selects one preferred positive example, including all of the negatives can lead to slow training. Many of the negatives may be redundant with other negatives or already be classified well by the model, suggesting that we can create comparable model quality more quickly by only including the most informative negatives. This paper a novel method for selecting negatives and supports it with theoretical analysis and empirical comparisons to recent benchmarks.

**Strengths:**

This paper studies an important problem, has a thorough theoretical analysis, and presents competitive empirical results against a number of recent baselines.

**Weaknesses:**

Based on my current understanding of the paper, the main weakness is that it selects a subset based on freezing all but the last layer and assumes that this selection will be effective when training the whole network. See Questions #1 and #4 below.

There are also some typos that limited understanding and my confidence in correctness. See questions #2, and #5.

**Questions:**

#1. My understanding of Assumption 4.1 is that we do not think about back-propagating gradients further back through the network architecture.  We only think about the effect of training the model weights in the last layer. Is this correct? If so, this seems like a major limitation of the work. At a minimum, readers should be made more aware of this limitation and ideally it would be analyzed.

In the numerical experiments, based on B.3, it seems that the whole network is trained. Is that correct?  This is good, if true.  More details should be added to B.3 and perhaps Section 6 to make this point more clear.

Empirically, it would be good to understand the gap between computing things with just the last layer vs. the whole network --- if one were to select subsets based on the log det of the regularized Fisher information matrix computed by backprogating through the full network, how different would the selected subsets be? Perhaps this is not computable at scale with an LLM, but it could be answered on some smaller-scale example.

#2. Either there is an important typo or I am confused about the paper.
On line 192-3 of page 4, n is given as the *total* number of negative responses.
On line 202-3 of page 4, it is stated that n is the *chosen* negatives.
The number of chosen negatives would typically be strictly smaller than the total number of available.
I read the rest of the paper assuming that n is the chosen number of negatives and that there is some unstated larger number of negatives from which these can be drawn.

#3.  The typo above made it hard to understand the following:

The notation H(S) in equation 11 uses Z_n -- this is held fixed during the greedy optimization proposed in lines 13 and 14.  I *think* this is the Z matrix that you get when you include *all* of the negatives. If it were the one you get when you just include the optimal ones, then it's unclear to me how you know it during the greedy procedure.

But if it is *all* of the data, then H(S^*_n) isn't the same thing as computing the log det of equation 10 with the selectetd negatives S^*_n. How do we know it is a good approximation? Presumably, the theory wants to hold sigma(Z_n) fixed during the greedy algorithm because it makes analysis simple.  But this seems to be another limitation of the method.

#4. Please walk me through how we get computational savings from selecting a strict subset of negatives.  The greedy algorithm requires us to compute v_j vectors for all of the negatives.  If the model actually satisfied Assumption 4.1, this seems like almost the same work as computing the gradient of the loss including all of the negatives. Does the savings come from the fact that we freeze everything but the last layer when we compute the v_j, but then when we actually do backpropagation, we compute through the whole network?

#5. In the high-probability bounds (Theorems 5.1 and 5.2), what is random?  I tried to figure this out by reading the proof of 5.1 in Appendix A.1 but it seems to rely heavily on Kveton et al. 2025 and uses notation that I didn't see defined elsewhere (Sigma_n, c_{min}). Also check the typo "covariant". Why does equation (26) follow with probability at least 1-delta?  And should the inequality above from Kveton et al. 2025 also be stated as holding only with high probability?

#6. The experiments state that the number of negatives was expanded from 4 to 20 for MedMCQA. Was this because the method provides little value when only 4 negatives are present?

---

> ### Author Response · Authors · 2025-11-24
> **Response to reviewer kyVT**
>
> Thank you for the thorough and constructive review. We appreciate the opportunity to clarify several important points.
>
> **Q1, Q4. Scope of Assumption 4.1, and where computational savings come from**
>
> Thank you for the careful questions. Your reading of Assumption 4.1 is correct in scope: to derive the Plackett Luce Fisher objective and finite sample bounds (Eqs. 10 to 12, Section 5), we linearize only the policy’s final layer and treat representations as frozen for the selection and analysis step. This is a standard tractability surrogate in recent RLHF and preference optimal design and neural linear bandits [1–5]. We will make this explicit in the main text. In experiments, we fine-tune the full LLM with all parameters updated, as described in Appendix B.3, and we will clarify this separation in B.3 and Section 6. Extending comparable Fisher-based guarantees to full transformers is an open theoretical problem. Most prior RLHF and preference optimal design theory [1-5] analyzes a frozen representation with a linear head because it makes Fisher or D-optimal objectives and finite sample bounds tractable, so we adopt the standard last layer surrogate for selection and theory.
>
> On computational savings, yes, the savings come from freezing everything but the last layer when computing $v_j$. The greedy step computes $v_j$ for all candidates using cached $d$-dimensional features, with no full model backprop. The savings then appear in fine-tuning: standard multi-negative DPO backpropagates through all $m$ negatives per step, while MASS-DPO backpropagates through only the selected $n \ll m$, giving an approximate $m/n$ reduction in backward cost and memory, with negligible one-time selection overhead.
>
> **Q2. Notation typo for $n$**
>
> Thank you for catching this typo. We will fix the typo in the final revision.
>
> **Q3. Notation typo:**
>
> Thank you for catching this. This is a notation typo: Eq. 11 is the general definition of the (regularized) Fisher information for an arbitrary subset, and the normalizer should be denoted $Z$ rather than $Z_n$. We will fix the typo in the final revision.
>
> **Q5.**
>
> Thank you for the careful questions. We will make the probability space explicit in Section 5. The randomness in Theorems 5.1 and 5.2 is over the sampled prompts and the stochastic Plackett Luce preference outcomes, hence the empirical loss, its gradient, and Hessian are random variables. In the final version, we will add the missing notation, i.e., $\Sigma_n$, which is the regularized information matrix, $\Sigma_n = \gamma I_d + \nabla^2 L_{\mathrm{DPO}}(\theta^\ast; S_n)$. We will likewise move the definition of $c_{\min}$ (currently in Assumption B.2) into Section 5, and fix the typo 'covariant' to 'covariance.'
>
> On the high probability qualifier: Eq. (26) is obtained by applying a standard self-normalized concentration bound to the stochastic gradient term in Theorem A.1, following the same argument as [1], with failure probability parameter $\delta$. Conditioning on this concentration event yields Eq. (26), and the draft already states that Eq. (26) holds with probability at least $1-\delta$. The intermediate inequality cited from [1] is part of the same concentration chain, so it is also a high probability statement under the same $1-\delta$ event. We will label that step explicitly in Appendix A.1 in the revision for clarity.
>
> **Q6. MedMCQA negatives**
>
> Yes, this was to make the selection nontrivial. MedMCQA provides one correct answer and three distractors; with only three negatives, there is no meaningful selection problem. As described in Appendix B.2, we expand each pool to 20 by pooling candidates within the same subject field. This allows us to evaluate whether MASS DPO can identify informative negatives in larger, realistic pools.
>
> ---
>
> We appreciate the reviewers’ constructive feedback and hope the clarifications above address all concerns.
>
> ---
>
> [1] Kveton, Branislav, Xintong Li, Julian McAuley, Ryan Rossi, Jingbo Shang, Junda Wu, and Tong Yu. "Active Learning for Direct Preference Optimization." arXiv preprint arXiv:2503.01076 (2025).
>
> [2] Das, Nirjhar, Souradip Chakraborty, Aldo Pacchiano, and Sayak Ray Chowdhury. "Active preference optimization for sample efficient rlhf." In Joint European Conference on Machine Learning and Knowledge Discovery in Databases, pp. 96-112. Cham: Springer Nature Switzerland, 2025.
>
> [3] Liu, Pangpang, Chengchun Shi, and Will Wei Sun. "Dual active learning for reinforcement learning from human feedback." arXiv preprint arXiv:2410.02504 (2024).
>
> [4] Scheid, Antoine, Etienne Boursier, Alain Durmus, Michael I. Jordan, Pierre Ménard, Eric Moulines, and Michal Valko. "Optimal design for reward modeling in rlhf." arXiv preprint arXiv:2410.17055 (2024).
>
> [5] Xu, Pan, Zheng Wen, Handong Zhao, and Quanquan Gu. "Neural contextual bandits with deep representation and shallow exploration." arXiv preprint arXiv:2012.01780 (2020).

---

### Meta-Review · Area_Chair_SyGe · 2026-01-08

**Summary:**

Paper studies selection of negative examples during DPO. Including all negatives can slow down training, so authors adapt the recently proposed D-Optimal Design framework for preference learning for sub-selecting the most-informative negatives. Paper studies an important problem and provides a more principled solution than prior wor. It has theoretical analysis and produce competitive results against recent baselines.

Unfortunately, the paper contains wrong theoretical claim and proof. Specifically for Lemma 4.3 was found be incorrect by Reviewer Reviewer VmBA and Reviewer P2ki. AC also concurs that both the statement and the proof (Lemma A.4 in Appendix) are incorrect. Paper also had many other typos and missing definitions and clarity in theoretical analysis.

**Reviewer Concerns:**

Authors argued that statement in Lemma 4.3 is incorrect in general it is not incorrect in their specific settings. They also argued that this result inconsequential to rest of the results of the paper. Unfortunately both the proof and the statement does not reflect this. Authors also did not attempt revise the paper to make the correct theoretical claim. Other issues were clarified or fixed during the rebuttal.

**Reviewer Scores:**

All the reviewers keeps their scores.

---

### Decision · Program_Chairs · 2026-01-26

Reject